# Subregional activity in the dentate gyrus is amplified during elevated cognitive demands

**Charlotte CM Castillon[1], Shintaro Otsuka[1], John N Armstrong[1], Anis Contractor[1,2,3]***

[1]Department of Neuroscience, Feinberg School of Medicine, Northwestern University, Chicago, United States; [2]Department of Psychiatry and Behavioral Sciences, Feinberg School of Medicine, Chicago, United States; [3]Department of Neurobiology, Weinberg College of Arts and Sciences, Northwestern University, Chicago, United States

## eLife Assessment

This manuscript presents a **valuable** study of the activity and functional relevance of different circuits in the dentate gyrus of mice performing a pattern separation task. **Convincing** evidence is presented to support the paper's central conclusions. The study is likely to be of interest to those studying the subregional organization and cell type-specific functions of the dentate gyrus.

**\*For correspondence:**
a-contractor@northwestern.edu

**Competing interest:** The authors declare that no competing interests exist.

**Abstract** Neural activity in the dentate gyrus (DG) supports the detection and discrimination of novelty, context, and patterns. Granule cell activation differs between the supra- and infrapyramidal blades across hippocampal-dependent tasks, yet how excitatory dynamics shape this blade-specific bias under varying cognitive demands remains unclear. Here, we combined an automated touchscreen pattern separation task in mice with temporally controlled tagging of active neurons to determine how increasing cognitive demand influences spatial activity patterns in the DG. As task difficulty increased, activation became progressively biased toward the suprapyramidal blade and was accompanied by structured distributions of active mature granule cells (mGCs) along both the radial and transverse axes. Selective inhibition of mGCs did not alter these spatial patterns, but profoundly impaired performance, as mice were no longer able to discriminate between closely spaced locations. In contrast, chemogenetic inhibition of adult-born dentate granule cells (abDGCs) beyond a critical maturation window impaired performance under high-demand conditions, increased overall mGC activity, and disrupted blade-specific organization even in animals that successfully completed the task. These findings demonstrate that high cognitive demand recruits spatially organized mGC activity and support a modulatory role for abDGCs in shaping dentate circuit dynamics.

## Introduction

The dentate gyrus (DG) is the input structure to the hippocampus and plays important roles in episodic memory (*Hainmueller and Bartos, 2020*), spatial coding (*Stefanini et al., 2020*), novelty detection (*Hunsaker et al., 2008*), as well as in separating patterns (*Leutgeb et al., 2007*). A long-standing proposal stemming from early computational studies suggests that the DG contributes to pattern separation based on its anatomical connectivity (*Marr, 1969*; *Marr, 1971*; *Rolls, 2016*). The DG is the gateway to the hippocampus, receiving direct excitatory input from the entorhinal cortex (EC). The

large number of granule neurons in the DG receiving input from a relatively smaller population of EC has been proposed to promote expansion recoding which, along with sparse firing of granule cells, allows the separation of inputs at the level of neural activity (*Borzello et al., 2023*).

The granule cell layer (GCL) of the DG is organized into two blades, termed the suprapyramidal (SB) and infrapyramidal (IB) blades. When viewed in transverse cross-section (*Amaral et al., 2007*), mature granule cells (mGCs) exhibit asymmetric activation across these blades during hippocampal-dependent tasks (*Chawla et al., 2005*; *Erwin et al., 2020*; *Snyder et al., 2009*; *Scharfman et al., 2002*), with greater activity typically observed in the SB compared to the IB (*Chawla et al., 2005*; *Erwin et al., 2020*; *Snyder et al., 2009*). However, the circuit-level mechanisms underlying this functional asymmetry remain unresolved.

In addition to mature populations, the subgranular zone (SGZ) of the DG serves as a neurogenic niche where new granule neurons continue to be generated throughout adulthood (*Altman and Das, 1965*; *Kaplan and Hinds, 1977*; *Cameron et al., 1993*; *Kempermann et al., 1997*). This process of adult neurogenesis gives rise to adult-born dentate granule cells (abDGCs), which can remodel local circuitry and potentially contribute to memory flexibility (*Fölsz et al., 2023*). Interestingly, the asymmetry in mGC activation across the blades parallels an asymmetry in neurogenesis, with higher numbers of abDGCs reported in the SB (*Dranovsky et al., 2011*; *Jinno, 2011*; *Ramirez-Amaya et al., 2006*; *Alves et al., 2018*).

In this study, we investigated whether performance in a cognitively demanding task alters the patterns of neural activity in the dorsal DG, and how manipulation of excitatory populations within the DG can modulate these activity patterns. Using TRAP2 mice (*DeNardo et al., 2019*) to label neurons activated during a touchscreen-based spatial discrimination task, we observed a consistent bias in activity between the two blades of the DG. When the cognitive demand of the task was elevated by reducing the degree of spatial separation, there was an increase of TRAP cells activity and a more prominent and amplified spatial bias with active neurons distributed preferentially to the SB and relatively reduced in the IB. To dissect the contribution of distinct excitatory populations, we selectively inhibited either mature or adult-born granule cells during task performance. We found that manipulating these populations had differential effects on the blade-specific bias of activation, indicating distinct roles for mGCs and abDGCs in shaping DG representations during spatial cognition. Inhibition of mGCs did not alter the blade bias of activation but markedly impaired performance, as mice were no longer able to discriminate closely spaced locations under high cognitive demand. In contrast, chemogenetic inhibition of abDGCs beyond a critical window of their maturation decreased task performance, increased overall mGC activity, and reduced the blade-specific bias of activation, suggesting that abDGC activity is required to maintain proper DG network organization under high cognitive load. Together, these findings reinforce the significance of neural activity in the dorsal DG in spatial pattern separation and highlight the circuit contributions of both abDGCs and mGCs to behaviors requiring a high cognitive demand.

## Results

### Amplified blade-biased activity of GCs during a high cognitive demand pattern separation task

Prior work has demonstrated that expression of immediate early genes in the DG including *Arc* (*Chawla et al., 2005*), *Fos* (*Snyder et al., 2009*), and *Egr1* (*Zif268*) (*Castillon et al., 2018*) correlate with behavioral state and are active in both mature and young abDGCs. Genetic strategies using TRAP1 mice (*Chatzi et al., 2019*) and TetTag mice (*Deng et al., 2013*; *Lamothe-Molina et al., 2022*) have also demonstrated that active GC populations can be labeled with these strategies. To determine whether behaviorally induced labeling of both mGCs and abDGCs is observed in TRAP2 mice (*Fos*[2a-iCreERT2];Ai9), we combined birth-dating of abDGCs with a single exposure to voluntary wheel running, which is known to activate DG neurons (*Chatzi et al., 2019*; *Figure 1—figure supplement 1A*). Mice given access to running wheels for 4 hr had elevated numbers of TRAP+ neurons throughout the DG, with both mature (mGCs; defined as granule cells not birth-dated with BrdU or EdU) and immature neurons labeled (*Figure 1—figure supplement 1B–F*). TRAP+ labeling of abDGCs was observed in birth-dated neurons 14 days post injection (dpi) from BrdU (*Figure 1—figure supplement 1E, F*). No double labeled TRAP+ BrdU+ neurons were observed in younger neurons prior to this timepoint

(*Figure 1—figure supplement 1F*). Importantly, an analysis of the blade-specific localization of TRAP+ neurons demonstrated that in home cage controls there was a small bias in the distribution of active neurons to the SB of the DG and this distribution of labeled GCs was not different in mice exposed to a single episode of running (*Figure 1—figure supplement 1D*). Moreover, consistent with previous reports (*Dranovsky et al., 2011*; *Jinno, 2011*; *Ramirez-Amaya et al., 2006*), we observed an asymmetry in neurogenesis, with a greater number of abDGCs in the SB compared to the IB (*Figure 1—figure supplement 1G*).

In order to determine the activation patterns of GCs (mGCs and abDGCs) when mice successfully performed a high cognitive demand spatial discrimination pattern separation task, TRAP2 mice were trained in a modified touch screen task that tests the ability to separate a unique pattern of illuminated squares posing a relatively low or high cognitive demand (*Figure 1A*, *Figure 1—figure supplement 2A, B*). The trial unique nonmatching-to-location (TUNL) task has been demonstrated to require the hippocampus, particularly when the cognitive demand is high, when the separation of the sample to test squares is reduced, or the delay between sample and choice is increased (*Talpos et al., 2010*; *Oomen et al., 2013*). Mice were injected with BrdU to birthdate neurons at day 0 (D0). At D7, mice were water restricted before EdU injections and shaping at D14 to habituate them to the operant task and learn to touch the screen for a saccharine water reward. Mice were then trained until they reached criterion (70% correct during one of two daily sessions on 2 consecutive days) in the TUNL task with a 1-s delay between the sample and choice and a large (L) separation between sample panel and the test panel (*Figure 1A*, *Figure 1—figure supplement 2A*). Mice reached criterion on average at 10.7 ± 0.8 days in the large separation sessions (*Figure 1B, D*, left panel). After reaching criterion, the mice were tested with alternating trials of large and small (LS) separation between the sample and test panels. In this case, the mice were always above criterion for the L trials but took another 9.3 ± 1.0 days to reach criterion on the S trials (*Figure 1C, D*, middle and right panels).

Different cohorts of mice were analyzed at each level of cognitive demand of the task to assess the activation of mGCs and abDGCs. TRAP labeling was used to identify active neurons in the DG, together with thymidine analogs that birth-dated abDGCs at approximately 3 weeks (25 ± 1.4 days, EdU) and 5 weeks (39 ± 1.2 days, BrdU) of age (*Figure 1E*). We found a strong relationship between the level of cognitive demand of the task and the degree of mGC activation, with a greater number of TRAP+ neurons in mice tested under higher cognitive load. Specifically, TRAP+ neuron density increased from shaping to the L separation stage and was highest in mice performing the most cognitively demanding condition (LS) (one-way ANOVA, $F(2,25)$ = 7.97, p = 0.0021; Tukey post hoc test. Groups Shap vs LS and Large vs LS were significantly different) (*Figure 1F*), regardless of whether the mice reached the criterion (C) or not (NC) (*Figure 1—figure supplement 2H*). A similar trend was observed when assessing c-Fos labeling 90 min after the final training session, further supporting the link between task demand and mGC activation (*Figure 1—figure supplement 2F*). Furthermore, when we examined the correlation between successful performance to criterion and the number of TRAP+ neurons, we found no significant relationship (*Figure 1—figure supplement 2C*), demonstrating that increased DG activity is driven by engagement in a high-demand task rather than task success.

Activation of adult-born granule cells (abDGCs) birth-dated with BrdU and EdU was assessed by colocalization with TRAP+ cells, rather than density, because the low number of newborn neurons precluded reliable density measurements. Accordingly, results are expressed as the proportion of birth-dated cells that were TRAP+. Using this approach, we did not observe a relationship between the level of cognitive demand and the recruitment of newborn neurons. Only a small fraction (~1%) of 25- to 39-day-old abDGCs were active (TRAP+) during the high cognitive demand task (LS) (*Figure 1G*), regardless of whether the mice reached criterion (C) or not (NC) (*Figure 1—figure supplement 2J*). Given the sparse labeling in this population, this percentage reflects very low absolute cell numbers per animal. In contrast, mice that performed only the low-demand version of the task (L configuration) exhibited almost no TRAP-labeled birth-dated cells (*Figure 1G*). Moreover, in the shaping group, no newborn neurons were co-labeled with TRAP, indicating that minimal cognitive engagement does not recruit this population (*Figure 1G*). Importantly, we did not detect qualitative differences in the overall abundance or blade distribution of BrdU+ or EdU+ neurons across behavioral groups (not shown).

In a separate cohort, we TRAPed neurons during the final day of LS trials and re-exposed mice to the same task 3 days later. After two LS sessions, mice were perfused 90 min later for immunohistochemical detection of endogenous c-Fos (*Figure 1A*). Task performance was reduced compared to

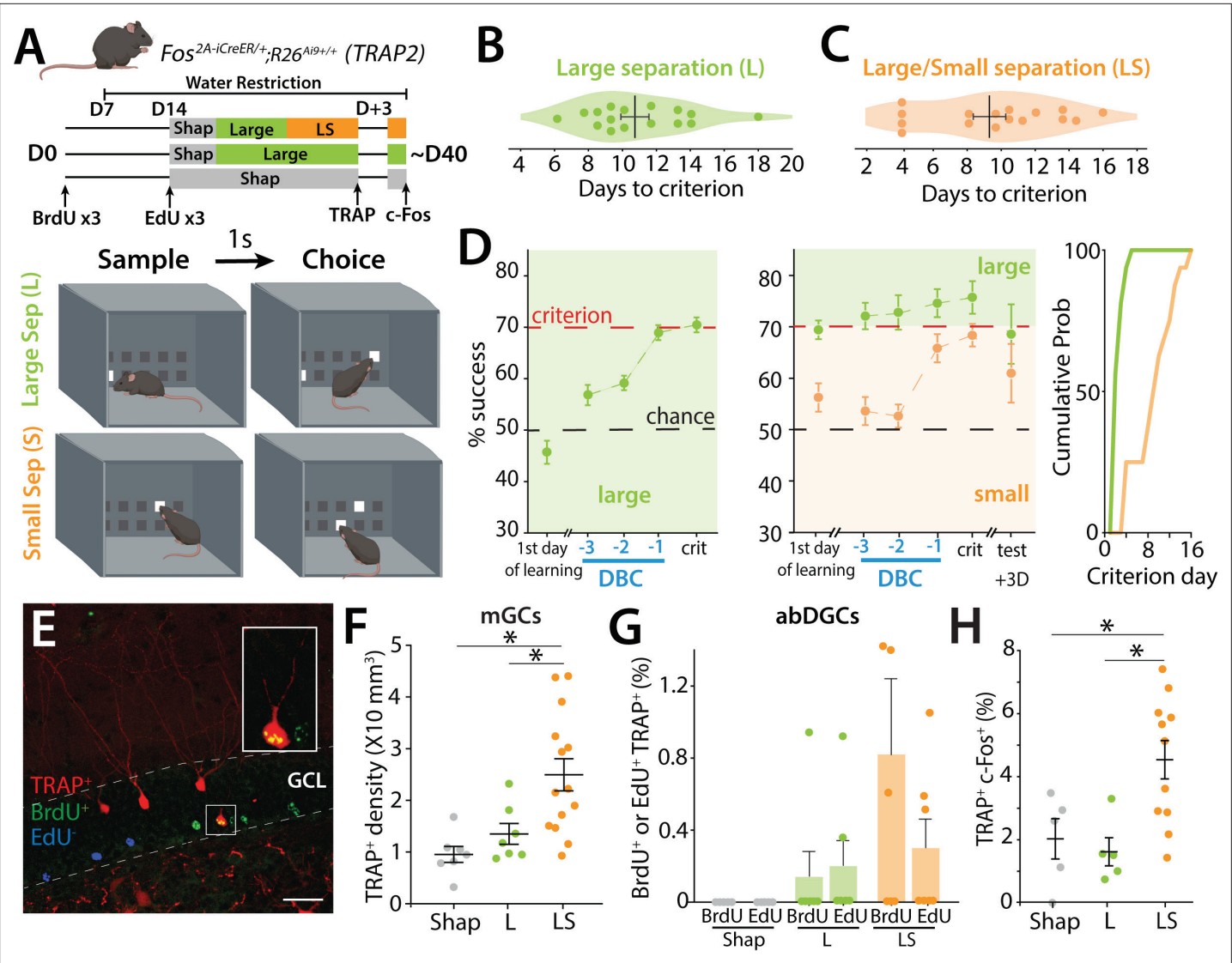

**Figure 1.** TRAP labeling of mGCs and birth-dated abDGCs after the TUNL pattern separation task. (**A**) *Top*: Schematic illustrating the pattern separation paradigm. Male (*n* = 9) and female (*n* = 7) mice (6- to 8-week-old) received three injections of BrdU at D0 and three injections of EdU 2 weeks later (D14) before starting the behavioral paradigm. Mice received one injection of 4-OHT on the day they reached criterion (TRAP) and were perfused 3 days later, 90 min after one exposure to the behavioral paradigm to allow endogenous c-Fos expression (c-Fos). Control mice (males *n* = 6, females *n* = 4) underwent the same treatment but only participated in the Large separation task (L). Mice were water-restricted during the behavioral experiment. *Bottom*: Cartoon representation of the chamber in TUNL task with examples of Large separation (L) and Small (S) separation configurations. High cognitive demand paradigm consisted of interleaved L and S trials (LS). (**B**) Number of days for each mouse to reach the 70% success criterion in the Large separation (L). (**C**) Number of days for mice to reach 70% success criterion in S trials, during the LS separation task. (**D**) *Left*: Average percentage of correct choices (% success) in the L separation on the first day of training, as well as during the 3 days prior to reaching criterion (days before criterion, DBC) and the day of reaching the criterion (crit) (one-sample *t*-test to chance at criterion Large *t*(16) = 12.04, p < 0.0001). *Middle:* Average percentage of correct choices in the L and S separation on the first day of training, during the 3 days before criterion (DBC), on the day of reaching criterion, and 3 days later (one-sample *t*-test to chance at criterion Large *t*(16) = 8.38, p < 0.0001; Small *t*(16) = 7.37, p < 0.0001). *Right*: Cumulative probability of days to criterion of mice in S for both controls L and LS groups. (**E**) Example section of BrdU+ labeling (green), EdU+ labeling (blue), and TRAP+ labeling (red) in TRAP2 mice. Calibration bar: 30 µm. (**F**) Density of TRAP-labeled GCs in mice undergoing LS compared to control mice engaged in either shaping (Shap) or the Large configuration task only (L) (one-way ANOVA, *F*(2,25) = 7.97, p = 0.0021; Tukey post hoc test. Groups Shap vs LS and Large vs LS were significantly different). (**G**) Percentage of BrdU labeled cells (39 ± 1.23 days) or EdU labeled cells (25 ± 1.37 days) also TRAP+ after L or LS training (BrdU+/TRAP+ L: 0.145 ± 0.145%, 6 mice; LS BrdU+/TRAP+ 0.822 ± 0.474%, 6 mice; ns; EdU+/TRAP+ L: 0.203 ± 0.144%, 6 mice; LS EdU+/TRAP+ 0.304 ± 0.155%, 7 mice; ns; Mann–Whitney). (**H**) Percentage of TRAP+ GCs expressing c-Fos+ in Shap, L, or LS task after reaching criterion (expressed as a percentage of TRAP+ cells) (one-way ANOVA, *F*(2,18) = 6.66, p = 0.0069; Tukey post hoc test. Groups Shap vs LS and Large vs LS were significantly different).

*Figure 1 continued on next page*

*Figure 1 continued*

The online version of this article includes the following figure supplement(s) for figure 1:

**Figure supplement 1.** Activity labeling of GCs after running.

**Figure supplement 2.** Description of the TUNL protocol and analysis of TRAP+ GCs in mice not meeting the criterion in the pattern separation task.

the final day of prior training in both the L and S trials (*Figure 1D*, middle panel). Analysis of mGC reactivation (TRAP+/c-Fos+ cells) revealed greater overlap in mice performing LS trials than in those performing only L separations or shaping sessions (one-way ANOVA, $F(2,18) = 6.66$, p = 0.0069; Tukey post hoc test. Groups Shap vs LS and Large vs LS were significantly different) (*Figure 1H*). Notably, across all animals and conditions, no BrdU+ or EdU+ cells were found to be labeled with both TRAP and c-Fos, indicating that newborn neurons activated during the final LS session were not reactivated 3 days later. Collectively, these results indicate that while increasing cognitive demand enhances activation and reactivation of mGCs, the recruitment of adult-born neurons remains low and does not scale with task difficulty.

Consistent with previous reports (*Chawla et al., 2005*; *Erwin et al., 2020*; *Snyder et al., 2009*; *Scharfman et al., 2002*), we observed a bias in granule cell activity within the DG, with TRAP+ mGCs preferentially localized to the suprapyramidal blade (SB). Interestingly, this bias was not fixed but varied according to the level of cognitive demand of the task (*Figure 2A*). Mice performing the high-demand LS condition showed a pronounced enrichment of TRAP+ mGCs in the SB compared to the infrapyramidal blade (IB), driven by both an increase in labeled cells in the SB and a decrease in the IB (two-way ANOVA (unbalanced) (blade: $F(1,38) = 146.34$, p = $1.33 \times 10^{-14}$) (group: $F(2,38) = 1.14 \times 10^{-7}$, p = 1) (group × blade: $F(2,38) = 13.99$, p = $2.81 \times 10^{-5}$), Tukey post hoc test, LS and Large (p = 0.03), LS and Shap (p = 0.05) for (SB and IB)) (*Figure 2A*, *Figure 2—figure supplement 1E*). In contrast, mice performing lower-demand versions of the task still exhibited a significant SB–IB difference, but the magnitude of this bias was reduced, resulting in a more balanced distribution of activity across blades. However, as with overall TRAP+ cell density, there was no significant correlation between individual task performance and the distribution of TRAP+ neurons in the SB (*Figure 1—figure supplement 2D*).

Further spatial mapping of TRAP+ cells relative to the SGZ and dentate apex revealed that active neurons were concentrated in the outer radial portion of the GCL—an area enriched in spatially tuned semilunar granule cells (*Erwin et al., 2020*; *Williams et al., 2007*), and this localization was particularly evident after the high-demand LS task (*Figure 2B, C*). In addition, active neurons were preferentially located toward the distal tip of the SB (two-way ANOVA (unbalanced) (distance (0–100%): $F(1,26) = 26.05$, p = $2.56 \times 10^{-5}$) (group: $F(2,26) = 0.044$, p = 0.957) group × distance (0–100%): $F(2,26) = 23.77$, p = $1.35 \times 10^{-6}$). Tukey post hoc test: Significant differences were observed between Shap; 0–50% and 50–100% (p < 0.0001), LS; 0–50% and 50–100% (p = 0.03) (*Figure 2C*). Together, these results demonstrate that the suprapyramidal bias in granule cell activity is dynamically modulated by cognitive demand, with the separation becoming more pronounced as task difficulty increases and producing distinct distributions of active neurons along both the apex–distal and hilar–outer axes of the DG (*Figure 2D, E*).

## Activation of GCs during remote recall in the high cognitive demand task

We next investigated whether a similar spatially biased activity of GCs was observed in a remote recall (R) test 3 weeks after reaching the criterion during the initial training (I), and whether there was overlap in the population of active DG neurons (*Figure 3A*). In the remote session, mice had a high success rate performing close to criterion in the L trials on the first day of retesting, suggesting that 3 weeks after initial learning they retained the ability in task (*Figure 3B*). However, when confronted with the more demanding trials with inclusion of the small separation configuration, mice required additional days to achieve the criterion level (*Figure 3C*), although most mice required less time to reach criterion in LS training during the remote trials than on the initial testing (*Figure 3D*). We again observed that the density of TRAP+ cells was increased in mice that underwent LS compared to L only trials during the initial training (not shown) and importantly the distribution of active neurons during both the initial training (TRAP+) and remote training (c-Fos+) were biased to the SB after LS training

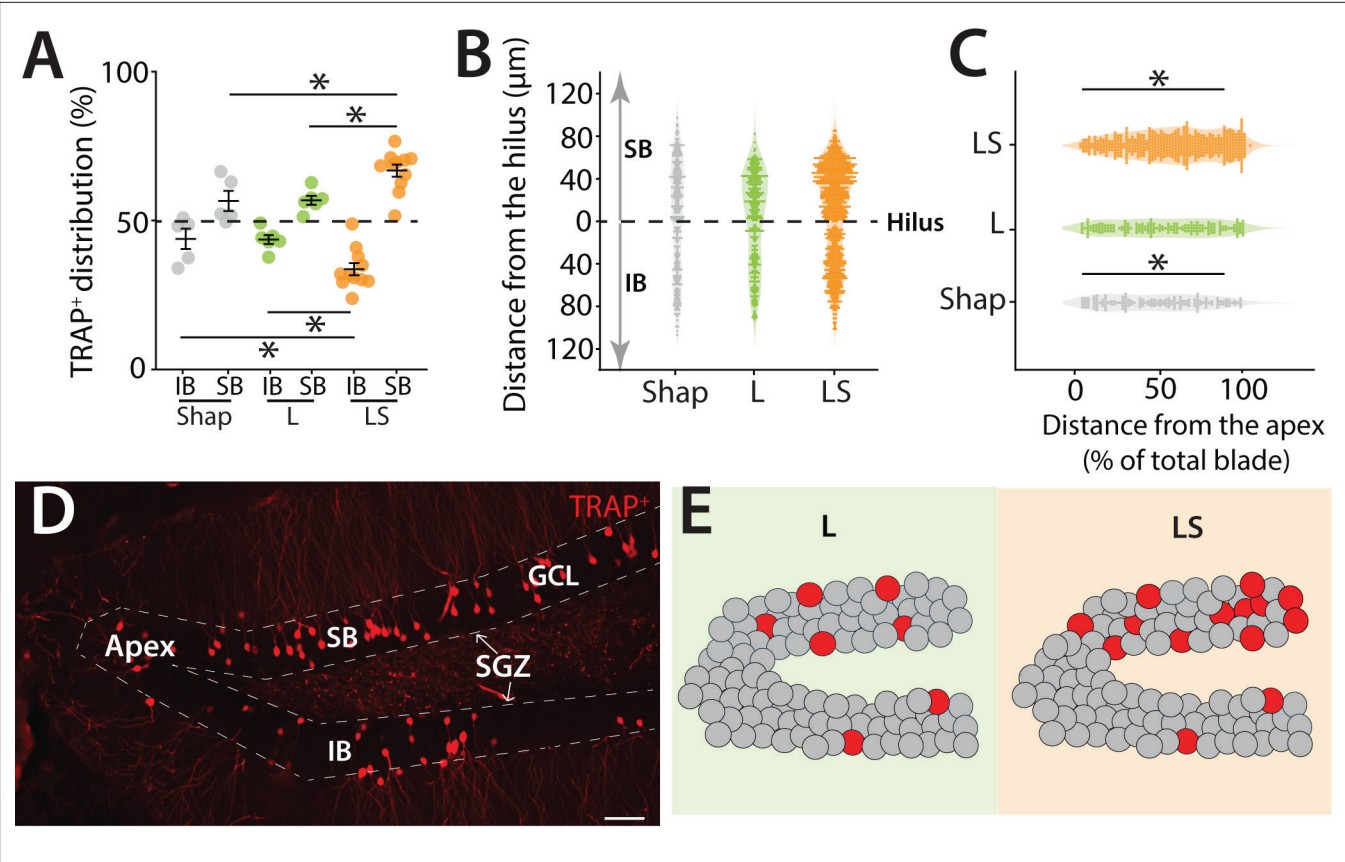

**Figure 2.** Blade-biased activity of mGCs during a high cognitive demand pattern separation task. (**A**) Spatial distribution of TRAP-labeled cells in the infrapyramidal (IB) and suprapyramidal (SB) blades of the dorsal DG in TRAP2 mice performing the LS separation, compared to control mice that performed only the Shap or L configuration two-way ANOVA (unbalanced) (blade: $F_{(1,38)} = 146.34$, p = $1.33 \times 10^{-14}$) (group: $F_{(2,38)} = 1.14 \times 10^{-7}$, p = 1) (group × blade: $F_{(2,38)} = 13.99$, p = $2.81 \times 10^{-5}$). Tukey post hoc test: significant differences were observed between IB, LS, and Large (p = 0.03), LS and Shap (p = 0.05), SB, LS, and Large (p = 0.03), LS and Shap (p = 0.05). (**B**) Spatial distribution of TRAP+ cells in the hilar to outer radial granule cell layer (GCL) axes in the SB (0–120 μm) and the IB (0–120 μm) of the dorsal DG with 0 μm indicating location at the hilar border (Shap: 5 mice, 255 cells; L: 5 mice, 294 cells; LS: 5 mice, 639 cells). (**C**) Distribution of TRAP-labeled cells along the apex to blade extremity axes of the SB of the dorsal DG, with 0% indicating location at the dentate apex (Shap: 5 mice, 152 cells; L: 5 mice, 176 cells; LS: 5 mice, 517 cells) (two-way ANOVA (unbalanced) (distance (0–100%): $F_{(1,26)} = 26.05$, p = $2.56 \times 10^{-5}$, group: $F_{(2,26)} = 0.044$, p = 0.957) (group × distance (0–100%): $F_{(2,26)} = 23.77$, p = $1.35 \times 10^{-6}$). Tukey post hoc test: significant differences were observed between Shap, 0–50% and 50–100% (p < 0.0001), LS, 0–50% and 50–100% (p = 0.03). (**D**) Example section of TRAP+ cells in the dorsal DG illustrating the distinct boundaries between the two blades, the subgranular zone (SGZ), and the apex. Calibration bar: 50 μm. (**E**) Cartoon illustrating the two blades of the DG: IB and SB. Dentate granule cells are depicted in gray while red circles represent TRAP+ cells in the DG. In the L group, a greater number of labeled cells are localized in the SB compared to the IB. This bias of activity is also observed in the LS group, where the bias is more pronounced, and the overall activity of the DG is increased. In the SB, the majority of the activated cells are located closer to the tip of the blade.

The online version of this article includes the following figure supplement(s) for figure 2:

**Figure supplement 1.** Dock10 is not expressed in DCX-positive immature newborn neurons but is expressed in Prox1-positive neurons.

(*Figure 3F, G*) (two-way ANOVA: blade: $F_{(1,12)} = 32.98$, p < 0.001; group: $F_{(1,12)} = 9.2 \times 10^{-30}$, p = 1; group × blade interaction, $F_{(1,12)} = 16.92$, p = 0.0014. Tukey post hoc test: significant differences were observed between IB; LS and Large (p = 0.05), SB; LS and Large (p = 0.05); *Figure 3G*: two-way ANOVA: blade: $F_{(1,12)} = 56.03$, p < 0.00001; group: $F_{(1,12)} = 0$, p = 1; group × blade interaction, $F_{(1,12)} = 7.02$, p = 0.021. Tukey post hoc test: no significant differences were observed between LS and Large for IB or SB).

An analysis of birth-dated active neurons labeled during the initial training (TRAP+) or during the remote trials (c-Fos+) demonstrated a small number of BrdU (~4 weeks) neurons were labeled during the initial training (*Figure 3H*) whereas in the remote training we observed both BrdU labeled (7.7 weeks) and EdU labeled (5.7 weeks) abDGCs that were c-Fos+ only in mice after LS training

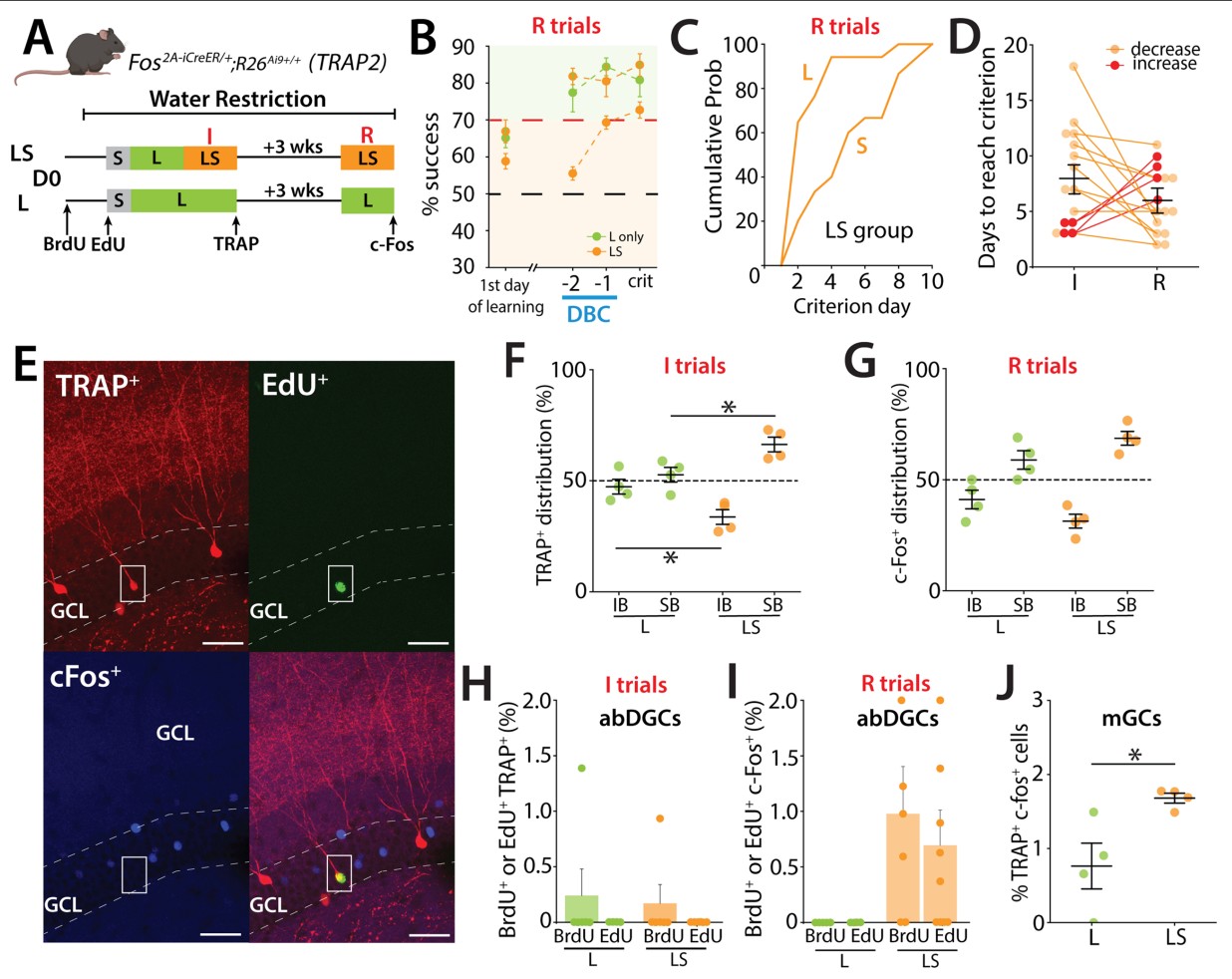

**Figure 3.** Activation of GCs during remote recall in the high cognitive demand task. (**A**) Schematic representation of the pattern separation paradigm used in the study. Mice (male *n* = 11, female *n* = 8) performed two training sessions: initial (I) and remote (R). The second training (R) took place 3 weeks after the mice reached the S criterion in LS. Mice were perfused when they reached the criterion in S during R. A second group that performed only in the L configuration during both training sessions was included as control (males *n* = 3, females *n* = 3). (**B**) Learning curves representing the percentage of success during the remote test (R). The control group trained only in the large-separation condition (L group) is shown in green and includes performance in the large-separation trials only. The LS group (orange) was trained in both large- and small-separation conditions and therefore includes two traces: the upper orange trace represents performance during large-separation trials, and the lower orange trace represents performance during small-separation trials. (one-sample *t*-test to chance at criterion; LS: Large *t*(14) = 11.8, p < 0.0001, Small *t*(14) = 10.3, p < 0.0001; L: Large *t*(3) = 6.78, p = 0.0066). (**C**) Percentage of mice reaching the L and S criterion in LS group, during the remote test (R) represented as cumulative probability. During the remote test, mice reach the L criterion in just a few days but require more time to reach the S criterion. (**D**) Performance of individual mice indicating a non-significant reduction in the number of days needed to reach the criterion between the initial and remote training sessions. Only 4 mice (red lines) required more days to reach the criterion during the remote test (I: 8.067 ± 1.173%, R: 5.93 ± 0.753, 15 mice; p = 0.204; Mann–Whitney). (**E**) EdU labeled cell that was also TRAP+ labeled after the initial training but was that was not reactivated (c-Fos) during the remote test. EdU (green), TRAP (red), and c-Fos (blue) in TRAP2 mice. Calibration bar, 10 μm. (**F**) Distribution of TRAP-labeled cells in the IB and SB blades of the dorsal DG in TRAP2 mice. Labeling was performed in LS separation during initial training (I) and compared to control mice that performed only the L configuration. Active neurons show a preferential distribution to the SB over the IB in the DG. This bias is more prominent in the LS group of mice two-way ANOVA: blade, $F(1,12) = 32.98$, p < 0.001; group, $F(1,12) = 9.2 \times 10^{-30}$, p = 1; group × blade interaction, $F(1,12) = 16.92$, p = 0.0014. Tukey post hoc test: significant differences were observed between IB; LS and Large (p = 0.05), SB; LS and Large (p = 0.05). (**G**) Distribution of c-Fos labeled cells between the IB and SB blades of the dorsal DG in TRAP2 mice performing the LS separation during remote training (R), compared to control mice that performed only the L configuration (two-way ANOVA: blade, $F(1,12) = 56.03$, p < 0.00001; group, $F(1,12) = 0$, p = 1; group × blade interaction, $F(1,12) = 7.02$, p = 0.021. Tukey post hoc test: no significant differences were observed between LS and Large for IB or SB). (**H**) Percentage of newborn neurons (BrdU (7.7 weeks) or EdU (5.7 weeks)) activated during the initial training (I), as indicated by co-labeling of TRAP+ (BrdU+ c-Fos+: L: 0.242 ± 0.242%, 6 mice, LS: 0.170 ± 0.170%, 6 mice; ns; Mann–Whitney). (**I**) Percentage of newborn neurons (BrdU or EdU) activated during the remote test (R), as indicated by c-Fos expression (BrdU+ c-Fos+: 0.977 ± 0.426%, 6 mice, p = 0.05; EdU+ c-Fos+: 0.694 ± 0.317%, 9 mice; p = 0.0785; Mann–Whitney). (**J**) Percentage of mGCs activated during the (**I**) training (TRAP+) that were reactivated during the R test (c-Fos+) (expressed as a percentage of TRAP+ cells) (unpaired *t*-test, *t*(6) = 2.90, p = 0.027).

(*Figure 3I*). No birth-dated abDGCs were found that were double labeled TRAP+ and c-Fos+; however, we observed dual labeled mGCs with more TRAP+/c-Fos+ labeled neurons in mice that had performed LS training than those that only performed L training (unpaired *t*-test, *t*(6) = 2.90, p = 0.027) (*Figure 3J*).

Taken together, these results demonstrate that during initial training and remote training in the pattern separation task, spatially biased activity of neurons is observed in the SB, but there is little overlap in activity of neurons in the DG of mice during these epochs of training.

## mGC activity is required for DG function and efficient pattern separation in TUNL

Given that the suprapyramidal bias in DG activity scaled with cognitive demand, we hypothesized that altering the excitatory drive within the DG might modulate this spatial pattern of activation. We began by targeting mGCs, the principal excitatory population in the DG, which are under strong inhibitory control and sparsely active in vivo (*Pilz et al., 2016*; *Danielson et al., 2016*).

To selectively suppress their activity, Dock10-Cre mice (*Kohara et al., 2014*) were crossed with R26-LSL-Gi-DREADD mice (designer receptor exclusively activated by designer drugs) to express the inhibitory DREADD hM4Di, which can be activated by the ligand deschloroclozapine (DCZ) (50 μg kg$^{-1}$) (*Nagai et al., 2020*) during the LS sessions of the task (*Figure 4C*). Characterization of Dock10-Cre expression in newborn neurons (BrdU+) revealed few labeled cells at 2 weeks postmitosis, with progressively more double-labeled abDGCs in older, non-DCX-positive neurons (*Figure 4A*, *Figure 2—figure supplement 1*), reaching ~80% of abDGCs by 7 weeks (*Figure 4A, B*). This pattern confirms that Dock10-Cre is expressed primarily in mGCs and older abDGCs. As predicted, DREADD-mediated inhibition of mGCs impaired task performance, with the majority of mice failing to reach criterion (62.5%) within the 14 day LS cutoff, compared to controls (34.8%) (p = 0.0452) (*Figure 4D, E*). Performance in the interleaved low-demand L trials was not affected (not shown). Consistent with reduced DG activity, we observed lower c-Fos expression in both the DG (unpaired *t*-test, *t*(9) = 2.30, p = 0.047) and CA3 (unpaired *t*-test, *t*(9) = 4.93, p = 0.0008) of mice that did not reach criterion after 14 days (*Figure 4F, G, I*). Interestingly, despite the overall reduction in activity, the distribution of active c-Fos+ neurons remained biased toward the SB, similar to control mice that reached criterion (*Figure 4H*) (two-way ANOVA unbalanced) (blade: *F*(1,16) = 315.83, p = 5.87 × 10$^{-12}$; group: *F*(1,16) = 0, p = 1; group × blade: *F*(1,16) = 0.040, p = 0.845). This suggests that the overall excitability is a key feature of DG function during performance of a high cognitive demand pattern separation task (*Figure 4H, J, K*).

## DREADD inhibition of abDGCs disrupts blade-biased activity of mGCs and performance of mice in a high cognitive demand task

Having established that mGCs regulate overall DG activity but do not determine the SB–IB bias, we next asked whether manipulating abDGCs could influence this bias and the pattern of DG activation. Because mGC inhibition reduced global excitability without altering the spatial organization of active cells, this suggested that the blade-biased pattern may be set upstream by a more selective cell population. Given that abDGCs constitute a smaller, more plastic excitatory population, we hypothesized that selectively altering their activity might differentially affect both overall excitability and the spatial distribution of active neurons during high cognitive demand tasks. abDGCs are known to affect the excitability of mGCs (*Ikrar et al., 2013*; *Drew et al., 2016*), and there is a demonstrated blade bias in neurogenesis in the DG. However, it remains unknown whether abDGCs affect the blade-biased activity patterns of mGCs that are evident during a pattern separation task. To acutely modulate the activity of abDGCs during task performance, we created mice in which the inhibitory DREADD (designer receptor exclusively activated by designer drugs) could be conditionally expressed in abDGCs by crossing R26-LSL-Gi-DREADD mice (hM4Di) (*Zhu et al., 2016*) to Ascl1$^{CreERT2}$ mice. This selectively expresses the hM4Di inhibitory DREADD in a birth-dated cohort of abDGCs by the delivery of TAM (tamoxifen, see Methods) in the diet of mice for 7 weeks allowing the inhibition of their activity with a selective ligand DCZ (50 μg kg$^{-1}$) (*Nagai et al., 2020*) during the LS sessions of the task (*Figure 5A, B*; *Yang et al., 2015*; *Tunc Ozcan et al., 2019*). Inhibition of abDGCs (≤7 weeks old) using the DREADD ligand DCZ increased the time animals took to reach criterion compared to control groups of animals that received DMSO (one-way ANOVA, *F*(2,31) = 3.77, p = 0.034; Tukey

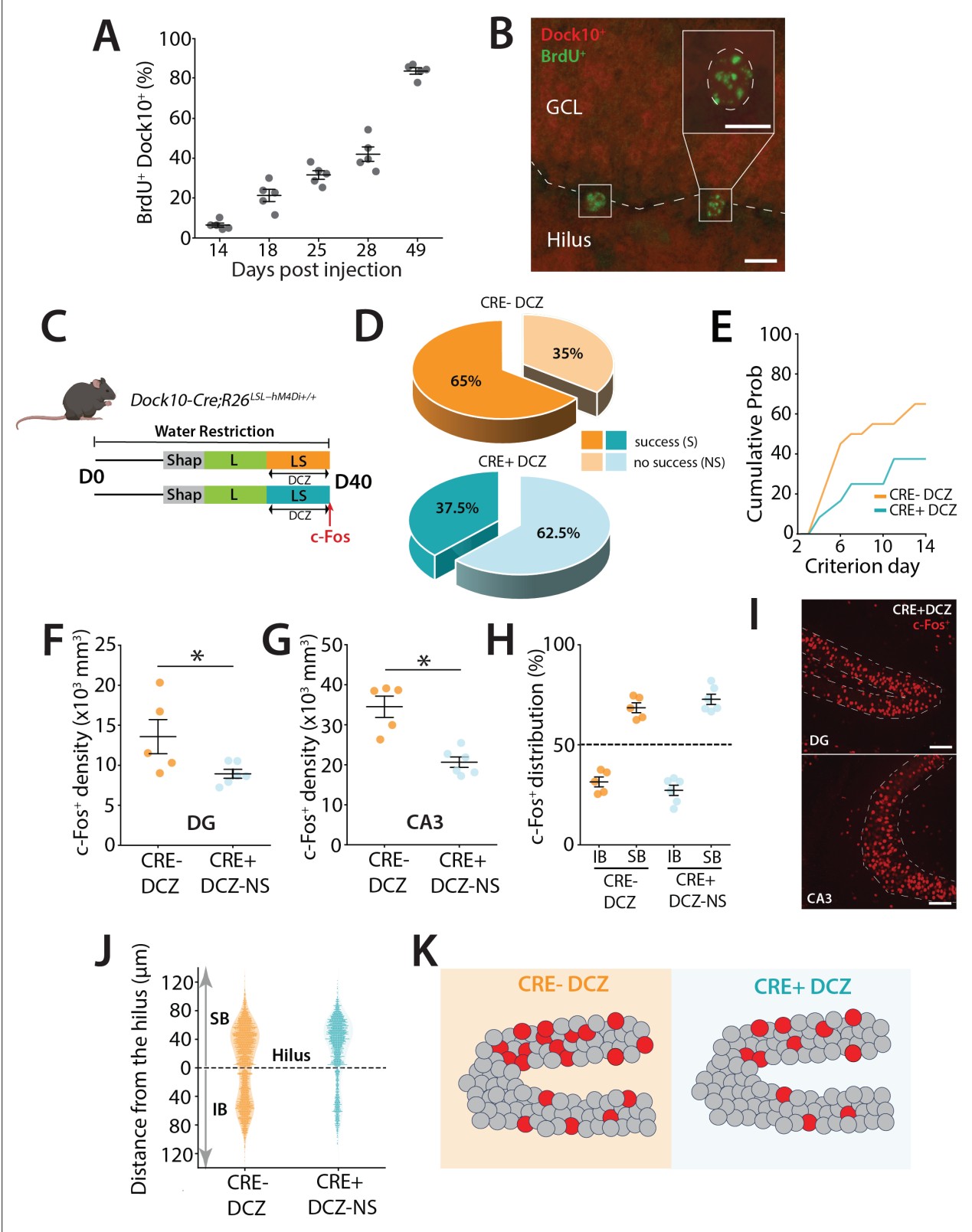

**Figure 4.** DREADD inhibition of mGCs during a high cognitive demand task. (**A**) Percentage of BrdU-labeled cells that are also Dock10+ at different time points following BrdU injections (in days). (**B**) Representative section showing BrdU+ labeling (green) and Dock10+ labeling (red) in Dock10 cre;Ai9 mice. Calibration bars: 10 μm. (**C**) Schematic representation of the pattern separation paradigm. All mice (males *n* = 17, females *n* = 27) received DCZ i.p. injections 30 min before the beginning of their LS trials. (**D**) Percentage of mice reaching criterion in (S) and percentage that were not successful

*Figure 4 continued on next page*

*Figure 4 continued*

(NS) after 14 days of training in S configuration of LS. A majority of mice never succeeded in reaching the criterion in the Cre+ DCZ group ($z$ = 2.00, p = 0.0453, $z$-test—Group NS). (**E**) Percentage of mice reaching the criterion in S trials across days represented as the cumulative probability including all successful and not successful mice in the group. (**F**) Density of c-Fos+ GCs in the dorsal DG across the two groups. A significantly reduced density was observed in the Cre+ DCZ group compared to controls (unpaired $t$-test, $t(9)$ = 2.30, p = 0.047). (**G**) Density of c-Fos+ GCs in the dorsal CA3 across the two groups. A significant reduction was observed in the Cre+ DCZ group compared to the control groups (unpaired $t$-test, $t(9)$ = 4.93, p = 0.0008). (**H**) Blade-specific distribution of c-Fos+ GCs in the IB and SB of the dorsal DG in mice performing LS separation. Two-way ANOVA (unbalanced, Type II) revealed a significant main effect of blade ($F(1,16)$ = 315.83, p = $5.87 \times 10^{-12}$), but no significant main effect of group ($F(1,16)$ = 0, p = 1) and no significant group × blade interaction ($F(1,16)$ = 0.040, p = 0.845). (**I**) Example of dorsal DG section from Dock10-cre:hM4Di mice after LS training. c-Fos+ neurons are labeled in red. Calibration bars: 100 μm. (**J**) Spatial distribution of c-Fos+ along the hilar to outer radial granule cell layer (GCL) axis of the DG (0–120 μm), with 0 μm indicating location at the subgranular zone (SGZ). (**K**) Cartoon illustrating the distribution of activity labeled GCs in the SB and IB blades of the DG. GCs are depicted in gray, while red circles represent activated cells in the DG. The overall distribution of neurons in mice in which mGCs were inhibited (Cre+ DCZ) and which did not reach criterion was not different from controls even though the total density of activity labeled neurons was lower.

post hoc test indicates that the Cre+ DCZ group is significantly different from the Cre+ DMSO, while other pairs are not significantly different) (*Figure 5C, D*). This impairment was specific to the cognitively more demanding LS task, as inhibiting ≤7-week-old abDGCs did not affect the performance of mice once they had learned the L configuration (not shown). Importantly, control groups that had also received TAM in their diet showed no effect in the pattern separation task or in the biased blade activity of mGCs (*Figure 5C–J*). Taken together, this experiment suggests that abDGCs in this cohort (≤7 weeks) are beneficial to performance in a pattern separation task.

Prior work has demonstrated that abDGC loss causes an increase in activity of mGCs (*Ikrar et al., 2013*; *Lacefield et al., 2012*) which is likely through a loss of connections to hilar interneurons which provide feedback inhibition to the DG. Consistent with this, analysis of c-Fos in the DG demonstrated increased labeling across the DG when abDGCs were inhibited with DCZ activation of hM4Di compared to control groups (p = 0.0271) (*Figure 5E, F*). Conversely, c-Fos+ labeling in the downstream CA3 region of the hippocampus demonstrated no significant difference in density between the control groups and abDGC inhibition (*Figure 5G*) (one-way ANOVA, $F(2,13)$ = 2.27, p = 0.143; Tukey post hoc test indicates that there is no significant difference between any pairs of groups). Interestingly, we found that the biased distribution of c-Fos+ neurons in the two control groups showed the same profile as we had observed before in the IB and SB after mice perform the high cognitive demand task (*Figure 5H*). However, in the cohort of mice in which abDGCs were inhibited, the prominent biased distribution pattern of activity was not observed (two-way ANOVA (unbalanced) (blade: $F(1,24)$ = 659.55, p < $2.2 \times 10^{-16}$; group: $F(2,24)$ = $3.48 \times 10^{-13}$, p = 1; group × blade: $F(1,16)$ = 0.040, p = 0.845). Tukey post hoc test. Significant differences were observed for IB comparisons: Cre+ DCZ vs Cre− DCZ (p = 0.000548) and Cre+ DCZ vs Cre+ DMSO (p = 0.00765). Significant differences were observed for SB comparisons: Cre+ DCZ vs Cre− DCZ (p = 0.000548) and Cre+ DCZ vs Cre+ DMSO (p = 0.00765)) (*Figure 5H*). Furthermore, the distribution of c-Fos+ cells within the blades showed that when abDGCs were inhibited during LS training there was a marked prominence of labeled neurons located near the hilus in both the IB and SB in the region closest to the SGZ, the neurogenic niche of the DG where abDGCs are produced, raising the possibility that these neurons are under greater feedforward inhibitory control via abDGCs (*Figure 5I*). In summary, DREADD inhibition of ≤7-week-old abDGCs during the LS high cognitive demand task resulted in a reduction in both in biased SB–IB spatial activity pattern as well as the activity labeling of neurons along both axes of the transverse DG (*Figure 5J*), and this altered activity correlated with a reduction in performance of animals which required increased days to complete the high cognitive demand task.

In the previous experiments we had provided TAM to mice in their diet to cause expression of hM4Di in neurons that range from 0 days to 7 weeks. This covers abDGCs that are within their critical period when they demonstrate strong plasticity and excitability and are well connected to the hippocampal network (*Ge et al., 2007*; *Kennedy et al., 2024*). Because inhibition of this 0- to 7-week population altered both performance and the blade-biased spatial distribution of mGC activity, we next asked whether these effects depend specifically on the maturity of the abDGCs. To test whether younger, ≤4-week-old neurons contribute similarly to DG function during high cognitive demand, we provided TAM for only 4 weeks to restrict hM4Di expression to these younger abDGCs (*Figure 6A*). In the LS high cognitive demand task, there was no difference in the time to criterion of animals that

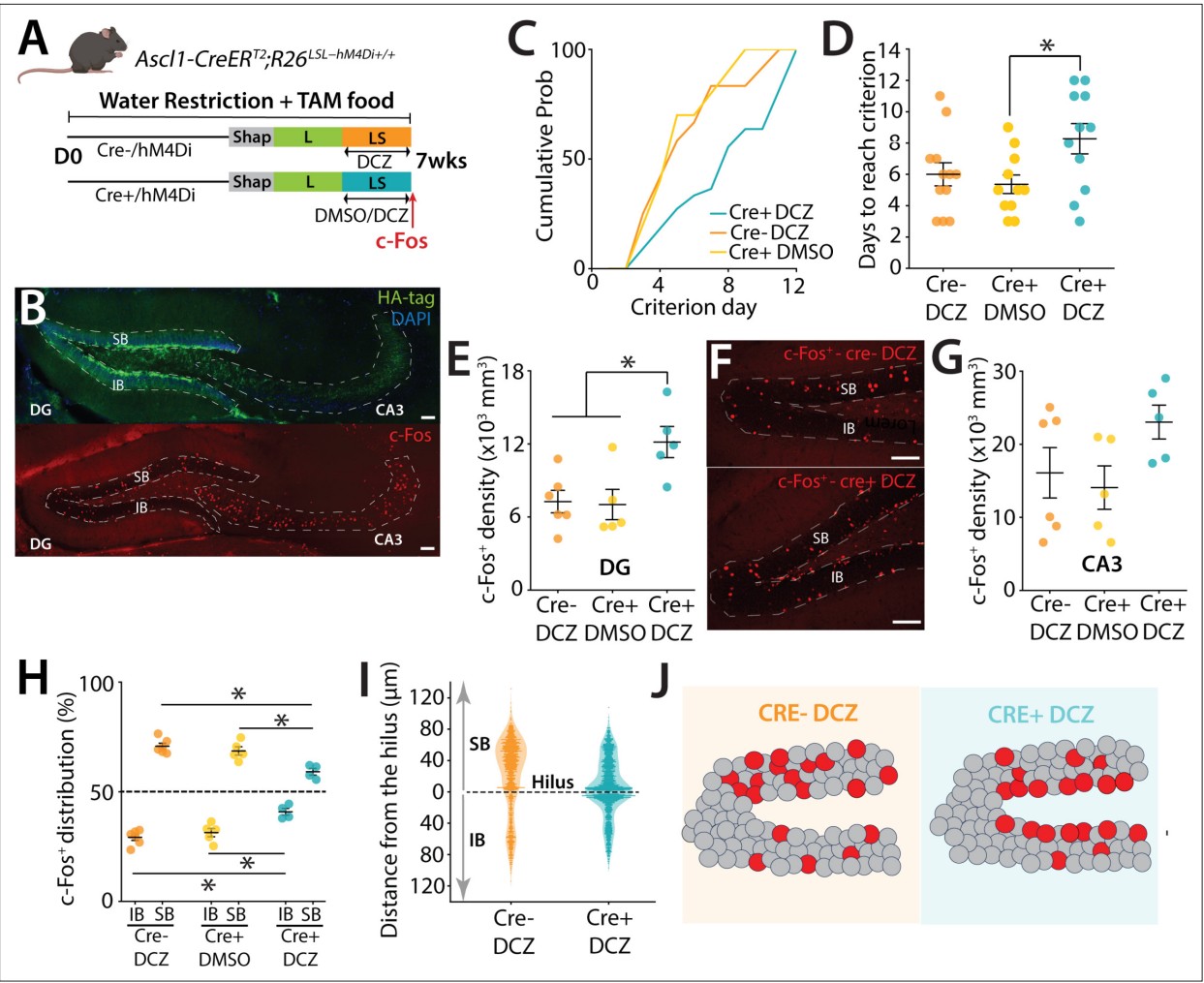

**Figure 5.** DREADD inhibition of ≤7 week abDGCs during a high cognitive demand task. (**A**) Schematic representation of the pattern separation paradigm. Prior to the onset of the training, mice (males $n$ = 17, females $n$ = 16) were provided tamoxifen-containing food ad libitum in the home cage for 7 weeks to activate TAM inducible Cre recombinase (Ascl1-CreER[T2]) in abDGCs. All mice received DCZ (deschlorochlozapine, 50 µg kg⁻¹) or DMSO injections 30 min before the beginning of LS trials. (**B**) Example sections from Ascl1-CreER[T2]; hM4Di mice with immunohistochemical localization of the HA tagged DREADD (top) and after LS training with c-Fos+ neurons labeled in red. Calibration bars: 140 µm. (**C**) Percentage of mice reaching the criterion in the S trials across days represented as the cumulative probability. The Cre+ DCZ treated group needed additional days to reach criterion compared to controls. (**D**) Number of days for each mouse to reach the 70% success criterion in S trials (one-way ANOVA, $F(2,31)$ = 3.77, p = 0.034; Tukey post hoc test indicates that the Cre+ DCZ group is significantly different from the Cre+ DMSO, while other pairs are not significantly different). (**E**) c-Fos+ cell density in all groups of mice. There was an increase in c-Fos+ mGCs in mice in which ≤7-week abDGCs were inhibited by DCZ ($H(2)$ = 7.22, p = 0.0271; Kruskal–Wallis) (Cre+ DCZ comparison to Cre+ DMSO, p = 0.014; Cre+ DCZ comparison to Cre- DCZ, p = 0.027 with Dunn's multiple comparison post hoc analysis). (**F**) Example section after c-Fos immunohistochemistry to assess the distribution of c-Fos+ mGCs between the IB and SB of the dorsal DG in groups of mice who received DCZ (cre− vs cre+). c-Fos cells are localized closer to the subgranular zone (SGZ) in the Cre+ group. Calibration bar, 100 µm. (**G**) c-Fos+ density in the CA3 region of the hippocampus in each of the groups of mice. No significant difference was observed in any of the groups (one-way ANOVA, $F(2,13)$ = 2.27, p = 0.143; Tukey post hoc test indicates that there is no significant difference between any pairs of groups). (**H**) Blade distribution of c-Fos+ mGCs in the IB and SB of the dorsal DG in mice performing the LS separation receiving either DCZ or DMSO 30 min prior to task performance. Active neurons are distributed preferentially to the SB than the IB in the DG for the control groups. However, this distribution bias is significantly reduced in the Cre+ DCZ group (two-way ANOVA (unbalanced) (blade: $F(1,24)$ = 659.55, p < 2.2 × 10⁻¹⁶; group: $F(2,24)$ = 3.48 × 10⁻¹³, p = 1; group × blade: $F(1,16)$ = 0.040, p = 0.845). Tukey post hoc test. Significant differences were observed for IB comparisons: Cre+ DCZ vs Cre− DCZ (p = 0.000548) and Cre+ DCZ vs Cre+ DMSO (p = 0.00765). Significant differences were observed for SB comparisons: Cre+ DCZ vs Cre− DCZ (p = 0.000548) and Cre+ DCZ vs Cre+ DMSO (p = 0.00765)). (**I**) Spatial distribution of c-Fos+ GCs along hilar to molecular layer axes of the SB (0–120 µm) and the IB (0–120 µm) of the dorsal DG, with 0 µm indicating the hilar position. c-Fos+ cells are localized closer to the SGZ in the Cre+ DCZ group. (**J**) Cartoon illustrating the DG with its two blades: IB and SB. Dentate granule cells are depicted in gray, while red circles represent activity labeled cells in the DG. For Cre+ DCZ mice, the distribution of labeled neurons is closer to SGZ, away from outer radial granule cell layer (GCL) and more evenly distributed in IB and SB.

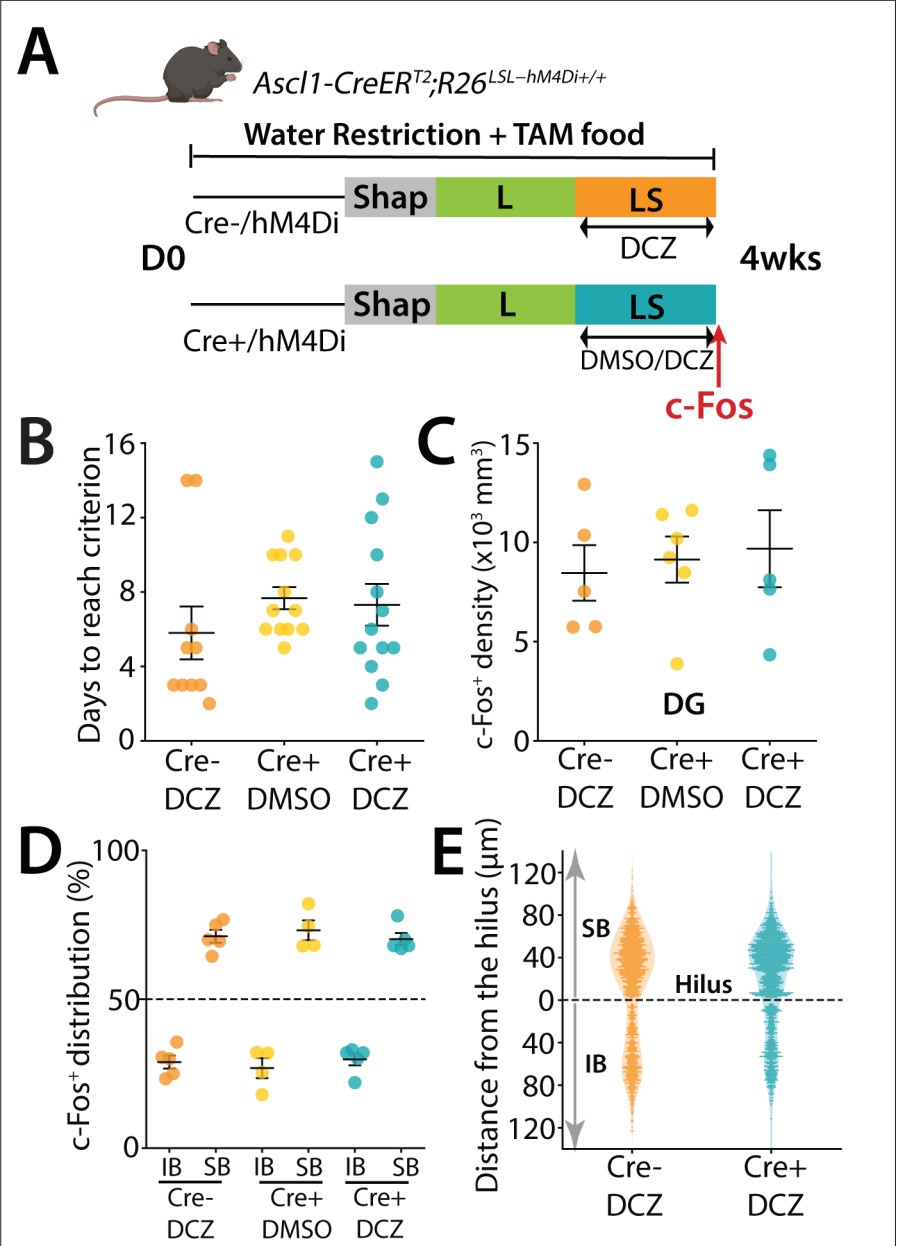

**Figure 6.** DREADD inhibition of ≤4-week-old abDGCs does not affect performance of mice. (**A**) Schematic representation of the pattern separation paradigm used in the study. Prior to the behavioral experiment, mice (males $n$ = 12, females $n$ = 23) were given tamoxifen-enriched food for 4 weeks to activate the Cre recombinase and label newborn neurons. All mice received DCZ or DMSO injections 30 min before the beginning of LS trials. (**B**) Number of days for each mouse to reach the 70% success criterion in S trials ($H$(2) = 4.21, ns; Kruskal–Wallis). (**C**) Density of c-Fos+ GCs in the dorsal DG across the different groups (one-way ANOVA, $F$(2,13) = 0.157, p = 0.857). (**D**) Blade-specific distribution of c-Fos+ cells in the IB and SB of the dorsal DG in mice performing LS separation ($H$(2) = 0.46, ns; Kruskal–Wallis). (**E**) Spatial distribution of c-Fos+ along the hilar to outer radial granule cell layer (GCL) axis of the DG (0–120 µm), with 0 µm indicating location at the border of the subgranular zone (SGZ) and hilus.

had received DCZ administration for hM4Di inhibition of abDGCs compared to the control groups (*Figure 6B*). Similarly, analysis of c-Fos+ density in the DG found no difference in overall density (*Figure 6C*) (one-way ANOVA, $F$(2,13) = 0.157, p = 0.857) and the blade distribution of active mGCs after hM4Di (*Figure 6D*) or in the hilar to molecular layer axes distribution (*Figure 6E*).

Taken together these results demonstrate that ≤7-week-old cohort of abDGCs that include neurons in their critical period of development influence the excitability and spatially biased pattern of activity of mGCs in the DG and also affect the performance of mice in a high cognitive demand pattern separation task. In contrast, young neurons at ≤4 weeks do not influence either the performance of mice in the task or the spatial distribution of activity in the DG.

## Discussion

### Blade-specific activation patterns in the dorsal DG during high-demand discrimination

In this study we used activity labeling techniques combined with a complex, automated visual spatial pattern separation task to capture patterns of activity of GCs in the dorsal DG. The DG is a critical part of the hippocampal formation that receives extrinsic input from cortex and other structures but is also highly regulated by local connections with hilar and CA3 neurons (*Borzello et al., 2023*). While many roles have been ascribed to the dorsal DG, a prominent theory has been that its activity is vital to pattern separation (*Guzman et al., 2021*). We found that activity labeling of neurons demonstrated unique spatial distributions of active GCs which were amplified when the separation task was made more stringent for the mice. We characterized this increased difficulty in the task as an increase in cognitive demand as it reflected the increased difficulty associated with discriminating highly similar spatial locations. However, the change in the task parameters likely engage a complex multitude of behavioral processes, including attention, motivation, reward expectation, and engagement. Therefore, our findings should be interpreted as reflecting activity patterns in the DG in response to increased task difficulty rather than a single isolated cognitive operation.

In addition, TRAP and c-Fos labeling integrate neuronal activity across a temporal window, these approaches primarily capture overall task engagement rather than activity linked to specific trial types or behavioral epochs. Accordingly, our findings should be interpreted as reflecting sustained neural recruitment during task performance rather than moment-to-moment encoding of individual trial events. In particular, we observed a biased distribution of active neurons in the SB, which was elevated in mice that completed the LS task. There have been prior descriptions of a blade-biased distribution of active neurons when mice engage in hippocampal dependent behaviors (*Chawla et al., 2005*; *Erwin et al., 2020*; *Snyder et al., 2009*; *Scharfman et al., 2002*); however, it is still not known how these spatially defined activity patterns are modulated or established. We found that not only is there a bias toward more activity in SB and less in the IB during a more demanding task, there is also an increase in activity density in the outer radial areas of the GCL where semilunar granule cells are enriched (*Erwin et al., 2020*; *Williams et al., 2007*), and in regions closer to the dentate apex. These elevated biased spatial distributions of active neurons were not observed when mice were assessed after running or after the low cognitive demand version of the task, two behaviors that increase the activity of mGCs significantly but do not enhance their blade-biased distribution (*Figure 1—figure supplement 2E*). However, while blade-specific differences in activity are evident, no distinct, independent microcircuit has been identified that exclusively governs the function of either the SB or IB. This raises an important question as to whether these differences arise solely from variations in afferent connectivity and intrinsic GC properties or if they reflect an underlying, segregated network organization within the DG. Future studies exploring whether the SB and IB operate as distinct computational units or as interconnected components of a larger network will be essential to clarify the functional basis of this observed bias.

### Activity level of mGCs is required for a high demand task

mGCs constitute the primary excitatory population of the DG and are essential for its sparse coding properties. Despite their large number, mGCs are strongly regulated by local inhibition and typically remain sparsely active in vivo (*Pilz et al., 2016*; *Danielson et al., 2016*). In our study, we used an acute and reversible DREADD inhibition of mGCs using a Cre driver line expressed in mature but not in young adult-born neurons (*Figure 4A*, *Figure 2—figure supplement 1*). Suppressing mGC activity during the LS task had a major effect on performance: most mice in the inhibited cohort failed to reach criterion. Analysis of behavior revealed that during S trials, performance remained at chance, whereas

in interleaved L trials, mice continued to perform above criterion. Consistent with reduced mGC excitability, the density of c-Fos+ neurons was significantly decreased in the DG.

Interestingly, despite this reduction in overall activation, the blade-biased distribution of active neurons was still evident in mice that performed the task but failed to reach criterion. Thus, unlike the effects observed when inhibiting adult-born neurons, there was no clear correlation between behavioral impairment and changes in the spatial distribution of active cells. Direct inhibition of mGCs therefore produced a fundamentally different outcome on DG circuit dynamics and task performance. While abDGC inhibition slowed performance and altered the spatial distribution of active mGCs even in mice that reached criterion, direct suppression of mGCs reduced global excitability without changing the spatial organization of active cells in animals that did or did not reach criterion (not shown). Moreover, the subset of mice that did reach criterion performed with latencies indistinguishable from control cohorts.

Prior studies using genetic manipulations to increase neurogenesis have identified blade-specific changes in the number of c-Fos+ cells, with an increase in the IB following contextual discrimination (*Besnard and Sahay, 2021*), and a corresponding increase or decrease in the IB or SB, respectively, after navigational learning (*Berdugo-Vega et al., 2021*). These prior findings support the hypothesis that the SB is context-specific and involved in pattern separation, while the IB is more associated with spatial representation and memory precision (*Berdugo-Vega et al., 2023*). Our study extends these observations by demonstrating that while overall mGC excitability is required for successful performance in high cognitive demand conditions, the SB–IB bias itself can persist independently of excitatory drive, suggesting that it is not directly determined by mGC activity.

## Adult-born granule cells modulate DG network organization during high cognitive demand

The DG, uniquely among temporal lobe structures, contains a neurogenic niche where new neurons are continuously generated and integrated into the existing network. Post-mitotic abDGCs mature over several weeks (*van Praag et al., 2002*), reaching maturity by around 8 weeks post mitosis where their functional properties and connectivity resemble those of other mGCs (*Zhao et al., 2006*; *Toni et al., 2007*). During a critical period between 4 and 8 weeks, however, they exhibit heightened plasticity and distinct circuit interactions (*Temprana et al., 2015*), making them uniquely positioned to influence local computation.

It has been proposed that abDGCs modulate local circuit activity by contributing to the sparsity of mGC firing through feedback inhibition (*Temprana et al., 2015*; *Johnston et al., 2016*; *Dieni et al., 2019*) or through direct inhibition (*Luna et al., 2019*). This inhibitory influence is largely mediated by diverse interneuron populations in the DG, including parvalbumin-positive (PV+), somatostatin-positive (SOM+), and neuropeptide Y-positive (NPY+) interneurons, which play crucial roles in regulating the excitability of GCs. Given their importance in enforcing sparse coding, these interneurons could be key modulators of the SB–IB activity differences observed in our study, yet their precise contribution to blade-specific activation remains unclear.

Consistent with prior behavioral work implicating the DG in pattern separation (*Tuncdemir et al., 2019*), manipulations that ablate abDGCs impair discrimination when task difficulty increases (*Clelland et al., 2009*), whereas genetic or environmental manipulations that enhance neurogenesis improve discrimination performance (*Sahay et al., 2011*; *Creer et al., 2010*). In our study, only limited numbers of abDGCs were labeled during task performance, as expected given their low population size relative to mGCs. However, neurons that were approximately 7 weeks post-mitosis were more prominently labeled than those 4 weeks post-mitosis, consistent with the notion that older abDGCs are more integrated into the local circuit. Despite their small number, these neurons exerted a disproportionate influence on DG activity.

Using an acute, reversible chemogenetic inhibition of different abDGC cohorts during task performance (*Tunc Ozcan et al., 2019*), we observed that suppression of ≤7-week-old abDGCs increased mGC activity, consistent with prior studies that chronically reduced neurogenesis (*Ikrar et al., 2013*). More importantly, inhibition of abDGCs during the LS task impaired performance in high-demand discrimination trials without affecting already-learned, low-demand trials within the same session. This manipulation disrupted the normal SB-biased pattern of mGC activation seen in controls, as well as the typical distribution of labeled mGCs along the hilar–molecular layer and apex–tip axes of the

DG. These effects were specific to the inhibition of ≤7-week-old abDGCs and were not observed when younger (≤4-week-old) or older abDGC populations were targeted. These findings suggest that abDGCs exert their influence not through direct encoding, but through regulating inhibitory tone within the DG microcircuit, potentially via PV- or SOM-positive interneurons that shape granule cell recruitment.

Notably, our results indicate that behavioral performance is not determined solely by the magnitude of DG activity. Inhibition of abDGCs increased overall mGC activation while disrupting the spatially organized, blade-biased pattern of activity, whereas direct inhibition of mGCs reduced overall activity without necessarily disrupting spatial organization in animals that remained capable of performing the task. These findings suggest that DG function depends on the coordinated recruitment and composition of granule cell ensembles rather than on total activity levels alone. Distinct perturbations may therefore impair behavior by altering the identity, balance, and coordination of neurons participating in task-relevant ensembles.

Together, these results reveal complementary but distinct contributions of mGCs and abDGCs to DG function. mGCs provide the essential excitatory output required for encoding and task performance under high cognitive demand, whereas abDGCs modulate this activity to establish the structured, blade-biased distribution of mGC activation necessary for optimal discrimination. This modulation likely arises through transient inhibitory control during the critical period of abDGC maturation, allowing the DG to dynamically tune its excitability and spatial coding to match task difficulty. Our findings support a model in which efficient pattern separation relies on both the stable excitatory scaffold provided by mGCs and the adaptive, modulatory influence of maturing abDGCs on DG network organization. This aligns with and extends prior models by showing that it is the maturing cohort, not immature or fully mature abDGCs—that provides this modulatory control.

## Limitations of the study

The use of activity tagging has the inherent limitation that it provides a snapshot of activity integrated across a temporal window and therefore cannot resolve which specific trial types, behavioral epochs, or decision processes drive neuronal recruitment. This is countered by the less invasive nature of these experiments than ones that require implants for in vivo recording of neuron activity, which are themselves not fit for purpose because they cannot capture the spatial distribution in the subregions of the DG effectively. Future studies using in vivo measurements of activity will be required to support the present findings and find conclusive neural representations required for discrimination of similar patterns.

Additionally, the experimental design in DREADD experiments required extended duration of tamoxifen administration which resulted in broad labeling of adult-born granule cells spanning multiple stages of maturation (approximately 0–4 and 0–7 weeks old), rather than a precisely birth-dated population. Consequently, the chemogenetic manipulations reflect the combined influence of a heterogeneous cohort of immature neurons, and do not resolve more finely the age-specific contributions of adult-born granule cells. Finally, the circuit mechanisms underlying blade-biased activity remain unknown, and future studies will be required to determine how these spatial patterns arise and how they contribute to computations within the DG.

## Materials and methods

**Key resources table**

| Reagent type (species) or resource | Designation | Source or reference | Identifiers | Additional information |
|---|---|---|---|---|
| Strain, strain background (*Mus musculus*) | Fos2A-iCreER/+ (TRAP2) | Jackson Laboratory | JAX 030323 | Used for activity dependent tagging |
| Strain, strain background (*Mus musculus*) | Ai9(Rosa-CAG-LSL-tdTomato-WPRE) | Jackson Laboratory | JAX 007909 | Cre-dependent reporter line |
| Strain, strain background (*Mus musculus*) | R26-LSL-Gi-DREADD (hM4Di) | Jackson Laboratory | JAX 026219 | Chemogenetic inhibition model |

*Continued on next page*

*Continued*

| Reagent type (species) or resource | Designation | Source or reference | Identifiers | Additional information |
|---|---|---|---|---|
| Strain, strain background (*Mus musculus*) | Ascl1-CreERT2 | Jackson Laboratory | JAX 012882 | Tamoxifen-inducible Cre line |
| Strain, strain background (*Mus musculus*) | Dock10-Cre | Susumu Tonegawa—RIKEN-MIT Cambridge, Massachusetts, USA | | Described previously (*Kohara et al., 2014*) |
| Strain, strain background (*Mus musculus*) | TRAP2;Ai9 | This paper | | Double-transgenic chemogenetic line |
| Strain, strain background (*Mus musculus*) | Ascl1-CreERT2;R26-LSL-hM4Di | This paper | | Double-transgenic chemogenetic line |
| Strain, strain background (*Mus musculus*) | Dock10-Cre;R26-LSL-hM4Di | This paper | | Double-transgenic chemogenetic line |
| Strain, strain background (*Mus musculus*) | Dock10-Cre; Ai9 | This paper | | Double-transgenic chemogenetic line |
| Chemical compound, drug | 5-Bromo-2'-deoxyuridine (BrdU) | Sigma-Aldrich | B9285 | Used for birth-dating of cells |
| Chemical compound, drug | 5-Ethynyl-2'-deoxyuridine (EdU) | Sigma-Aldrich | 900584 | Thymidine analog for labeling dividing cells |
| Chemical compound, drug | 4-Hydroxytamoxifen (4-OHT) | Sigma-Aldrich | H7904 | Used for TRAP labeling |
| Chemical compound, drug | Tamoxifen diet | Envigo | TD.130858 | Used for CreERT2 induction |
| Chemical compound, drug | Deschloroclozapine (DCZ) | MedChemExpress | HY-42110 | hM4Di agonist |
| Chemical compound, drug | DNase I | Sigma-Aldrich | D5025 | Used for BrdU staining |
| Chemical compound, drug | Click-iT EdU Imaging Kit | Thermo Fisher Scientific | C10340 | For EdU detection |
| Antibody | Rabbit Monoclonal anti-c-Fos | Cell Signaling Technology | #2250 RRID:AB_2247211 | (1:1500) |
| Antibody | Sheep Polyclonal anti-BrdU | Novus Biologicals | NB500-235 RRID:AB_10079353 | (1:2500) |
| Antibody | Rabbit Polyclonal anti-RFP | Abcam | Ab124754 RRID:AB_10971665 | (1:5000) |
| Antibody | Rabbit Monoclonal anti-HA tag | Cell Signaling Technology | #3724 RRID:AB_1549585 | (1:500) |
| Antibody | Donkey anti-rabbit Alexa Fluor 488/647 | Invitrogen | A-21206 RRID:AB_2535792A-31573 RRID:AB_2536183 | (1:1000) |
| Antibody | Donkey anti-sheep Alexa Fluor 488 | Invitrogen | A-11015 RRID:AB_3750347 | (1:1000) |
| Other | ProLong Diamond Antifade Mountant with DAPI | Invitrogen | P36971 | |
| Other | Wireless running wheels | Med Associates | ENV-047 | Used for running paradigm |
| Other | Operant touchscreen chamber (Operant House) | This paper; *Otsuka et al., 2025* | | Custom-built behavioral apparatus |
| Software | Python | This paper | https://shintaro18.github.io/ | Task control scripts |
| Software | Fiji (ImageJ) | NIH | https://imagej.net/software/fiji/ | Image processing and analysis |

*Continued on next page*

*Continued*

| Reagent type (species) or resource | Designation | Source or reference | Identifiers | Additional information |
|---|---|---|---|---|
| Software | Neurolucida | MBF Bioscience | - | Anatomical tracing |
| Software | Zeiss LSM700 software | Zeiss | - | Confocal imaging acquisition |
| Software | Statview | SAS Institute | - | Statistical analysis |
| Software | Origin | OriginLab | | Statistical analysis and plotting |

## Ethics statement

All animal procedures were conducted in accordance with protocols approved by the Northwestern University Institutional Animal Care and Use Committee (IACUC). Male and female mice were housed 3–5 per cage on a reverse light/dark cycle of 12/12 hr within a controlled environment. They received food and water ad libitum. All behavioral testing was conducted during the animals' dark cycle and was conducted by experimenters blinded to genotype. Tail biopsies were collected for genotyping via PCR. All efforts were made to minimize the number of mice used and to reduce any potential pain or stress, in line with ethical guidelines.

## Animals

All mouse lines were maintained on a C57Bl/6 background. Mice were propagated by female breeders heterozygous for Cre recombinase and homozygous for either the Ai9 allele (Rosa-CAG-LSL-tdTomato-WPRE) or hM4Di allele (R26-LSL-Gi-DREADD) with male breeders homozygous for Ai9 or hM4Di. TRAP2;Ai9 mice were generated by crossing *Fos*$^{2A-iCreER/+}$ (TRAP2) mice (Jax #030323) (*DeNardo et al., 2019*) with Ai9 (Jax #007909) mice to generate double-heterozygous mice. Ascl1-CreER$^{T2}$;hM4Di mice were generated by crossing R26-LSL-Gi-DREADD mice (Jax #026219) (*Zhu et al., 2016*) with mice carrying tamoxifen-inducible Cre recombinase under the control of the *Ascl1* promoter (Jax #:012882) (*Kim et al., 2011*) to generate double-transgenic progeny Ascl1-CreER$^{T2}$;R26$^{LSL-hM4Di}$. Cre-negative littermates from heterozygote breedings were used as controls. Dock10-Cre;hM4Di mice were generated by crossing Dock10-Cre (*Kohara et al., 2014*) mice with hM4Di mice to produce double-transgenic progeny Dock10-Cre;R26$^{LSL-hM4Di}$.

## Birth-dating of abDGCs

Six- to eight-week-old mice received three intraperitoneal (i.p.) injections of 5-Bromo-2'-deoxyuridine (BrdU, Sigma-Aldrich #B9285) at 100 mg/kg (i.p) at 4 hr intervals on a single day (*Castillon et al., 2020*). Two weeks later, 5-ethynyl-2'-deoxyuridine (EdU, Sigma-Aldrich, #900584) was administrated following the same injection protocol. EdU was used as an additional thymidine analog to label a distinct cohort of dividing cells. Because BrdU and EdU are incorporated into DNA during S-phase following systemic administration, the birth-dating timeline and survival intervals applied to BrdU labeling also apply to EdU labeling.

In this study, mGCs are operationally defined as dentate granule cells that were not labeled with BrdU or EdU and therefore are not classified as adult-born neurons within the defined labeling window. While the majority of these cells are expected to represent developmentally generated mature neurons, we cannot exclude the possibility that a small fraction may include younger, unlabeled neurons generated outside the labeling period.

## Tamoxifen administration

*IP injections:* To time the labeling of TRAP+ cell in TRAP2 mice, 4-hydroxytamoxifen (4-OHT; Sigma-Aldrich, #H7904) was administered intraperitoneally. 4-OHT was dissolved at 20 mg/ml in pure ethanol and stored at –20°C for up to several weeks. Before use, 4-OHT was redissolved in corn oil (Sigma, Cat #C8267) and heated to 90°C with shaking until the ethanol had evaporated, yielding a final concentration of 10 mg/ml. The final 10 mg/ml 4-OHT solutions were used on the day it was prepared. All injections were delivered intraperitoneally at 50 mg/kg either at the end of the training phase of the day, or 2 hr after running.

*Oral administration*: For conditional expression of hM4Di in *Ascl1-CreER^(T2)* cells, a tamoxifen diet (TD.130858, Envigo) was provided ad libitum for 4 or 7 weeks, depending on the experimental timeline. Body weight was carefully monitored daily. The chow contained 500 mg tamoxifen/kg, delivering ~80 mg tamoxifen/kg body weight per day, assuming an average mouse weight of 20–25 g and a daily intake of 3–4 g.

Mice were monitored daily for signs of stress, discomfort, or adverse effects related to tamoxifen exposure.

## Inhibitory DREADD-hM4Di activation

DCZ (MedChemExpress #HY-42110) was dissolved in 2% DMSO in 0.9% saline, and administered by i.p. injections at a dose of 50 µg kg⁻¹ (*Nagai et al., 2020*). Behavioral trials started 30 min after drug administration.

## Behavioral procedures

Adult (8–10 weeks-old) male and female mice were used in behavioral studies. To acclimate the mice to handling before behavioral testing, all animals were handled by the experimenter for 2 min per day for 3 consecutive days.

## Running wheels

To habituate the mice to running, wireless running wheels (ENV-047, Med Associates) were placed in their home cage for 5 days (maximum 5 mice per cage). Two days prior to BrdU injections, the wheels were removed from the cage. One week before the experiment, mice were individually housed. On the day of the experiment, a running wheel was added to each cage for a duration of 4 hr. After 2 hr, mice received a 4-OHT injection and were returned to their respective cages. Following an additional 2 hr of running, the wheels were removed from the cages. Three days later, wheels were reintroduced to the cage and remained until mice were euthanized for assessment of TRAP+ cells, 90 min after the beginning of the run. The effect of exposure to the wheels was quantified by measuring the total distance run by each mouse.

## Automated touch screen operant behaviors

Testing was conducted in custom-built, touch screen-based automated operant device boxes designed specifically for mice, the Operant House (https://www.biorxiv.org/content/10.1101/2025.01.24.634815v1; *Otsuka et al., 2025*). The apparatus consisted of a rectangular modular testing chamber (*w*14.5 cm × *l*21.5 cm × *h*18.5 cm). A 60-mW green LED was located above a port that gave access to a retractable water bottle at the back of the chamber. At the opposite end of the chamber, a flat-screen infrared touch monitor was controlled by custom scripts (https://shintaro18.github.io/; *Otsuka, 2025*). In front of the touchscreen was a mask with two horizontal lines of five squares. Mice performed a modified version of the 'trial unique nonmatching-to-location' (TUNL) task (*Oomen et al., 2013*). Five days prior to testing and during the training phase, mice were placed on a water-restriction regime, with access to water for 10 min per day, and were maintained at 85–90% body weight during the entire duration of the experiment. Mice were trained each day, in two separate sessions of 30 trials or a maximum of 30 min.

*Training phases—Shaping*: Following 30 min of habituation to the operant chamber, mice were trained to touch a lit white square stimulus presented randomly in one of the response windows of the touch screen in order to receive a reward (2 s water nozzle access with saccharin water 0.01%). Mice were trained in Shaping until they successfully achieved 30 rewards within 30 min during three successive sessions.

*Training phases—Large-only separation (low cognitive demand)*: After shaping, mice were trained in the TUNL task with a Large-only separation (L configuration). For this task, each trial was composed of two phases. During the sample phase, a white square was illuminated in one of the 10 possible locations on the screen (*Figure 1*, *Figure 1—figure supplement 2A, B*). Following a nose-poke to the sample square, the stimulus disappeared, and the mouse received a reward only on 33% of the trials. After a 1-s delay from the sample trial, the choice trial was initiated, where two stimuli were presented: one in the repeated sample location (incorrect) and the other in the new location (correct), separated spatially by four squares in the L configuration (*Figure 1A*, *Figure 1—figure supplement*

2A, B). A touch to the correct sample location resulted in a reward and a 5-s inter-trial interval (ITI) before the next trial was initiated. An incorrect choice resulted in a 10-second punishment (roof lights ON) and then an ITI, followed by correction trials during the first 2 days of training. Correction trials followed the same procedures, except that the same sample and choice locations from the previous incorrect trial were presented again until the correct choice was made. The correction trials were not included in performance calculations. Mice progressed to the next training phase after completing ≥30 trials within 30 min and achieving >70% correct responses in at least two of four sessions across two consecutive days.

*Training phases—Large + Small separation (high cognitive demand)*: Mice were then trained in sessions containing both large-separation (L) and small-separation (S) trials (LS configuration). Training followed the same pattern as previously described, except that there were two separate possibilities for the presentation of the choice square, which was either separated by 4 squares from the sample square during L trials, or the choice square was 0–1 square separated from the sample location during S trials (*Figure 1A*, *Figure 1—figure supplement 2B*).

Each session consisted of 15 L and 15 S trials presented in alternating order. Mice were trained until reaching criterion defined as ≥70% correct responses on S trials in two of four sessions across two consecutive days.

Approximately 25–30% of mice across experimental groups did not reach LS criterion within the allotted training period (*Figure 1—figure supplement 2G*).

## Immunohistochemistry

Mice were deeply anesthetized with isoflurane and then perfused transcardially with PBS containing 0.02% sodium nitrite, followed by 2% paraformaldehyde (PFA) in 0.1 M sodium acetate buffer (pH 6.5) for 3 min, and 2% PFA in 0.1 M sodium borate buffer (pH 8.5) for 10 min (*Sloviter et al., 1996*). Brains were dissected and post-fixed in 2% PFA 0.1 M sodium borate buffer (pH 8.5) for 24–48 hr and stored at 4°C in PBS until sectioning. 50 µm floating sections were collected on a Leica Vibratome VT1000 and stored at 4°C in PBS containing 0.02% sodium azide. For c-Fos analysis, perfusions were performed 90 min after the end of the behavioral training. Immunohistochemistry was conducted at room temperature using standard procedures. Free-floating sections were incubated for 30 min in a solution of 25 mM glycine to decrease the background signal and then rinsed two times in Tris-Buffered Saline 1X (TBS). Sections were then blocked for 1 hr in the saturation solution containing TBS-Triton (0.25% Triton X-100 (Fisher, BP151-100), 3% normal donkey serum (Jackson ImmunoResearch, 017-000-121), and 0.1% bovine serum albumin (BSA; Sigma, A9085)). The sections were then incubated overnight in the saturation solution containing the primary antibody rabbit anti c-Fos (1/1500, Cell Signaling), sheep anti-BrdU (1/2500, NovusBio), rabbit anti RFP (1/5000, Abcam), rabbit anti HA-Tag (1:500, #3724, Cell Signaling). For c-Fos staining, no serum and BSA were used. For BrdU staining, sections were incubated for 10 min in a solution of DNAse I (Sigma #D5025, 15 KU/ml dissolved in 0.05 M NaCl), MgCl$_2$ (6 mM), and CaCl$_2$ (10 mM) to cleave DNA then rinsed 2 × 5 min in TBS before the saturation part. For EdU staining, sections were incubated in solution from the kit ClickIT (#C10340 Thermo Fisher) during 30 min then rinsed 2 × 5 min TBS before the saturation part. No antibodies were necessary for the rest of the immunostaining. The next day, sections were washed 2 × 5 min in TBS before being incubated for 1 hr in secondary antibody solution (donkey anti-rabbit Alexa 647 or 488 (1/1000, Invitrogen), donkey anti-sheep Alexa 488 (1/1000, Invitrogen)). Finally, sections were washed 2 × 5 min in TBS and cover-slipped with ProLong Diamond mounting media 275 containing DAPI (#P36962, Thermo Fisher Scientific).

## Imaging and neuron quantification

Analyses were conducted in the dorsal DG, as prior studies have demonstrated its involvement in spatial memory and the fine-grained contextual differentiation required for pattern separation. The dorsal hippocampus is crucial for encoding and distinguishing spatial information, which is central to this process. In contrast, the ventral DG is more strongly associated with emotional processing and affective memory rather than high-resolution spatial encoding needed for pattern separation.

Confocal images were obtained with a Zeiss microscope (LSM700) with a 20x objective with the experimenter blind to the genotype and condition following previous protocols (*Castillon et al., 2020*). One section in every ten serial sections was analyzed through the dorsal hippocampus (stereotaxic

coordinates: –1.20 to –2.30 relative to bregma according to Paxinos' brain atlas). Two to three 10 µm z-stack acquisitions were made to picture the whole DG in each section. 3D reconstructions were used to univocally verify co-localization (e.g. activated mature neurons or immature neurons). All acquisitions were carried out in sequential scanning mode to prevent cross-bleeding between channels. Stacks of images were then reconstituted using Fiji software, allowing the experimenter to count the number of cells. The surface area of the dorsal hippocampus, DG, SB, IB, and CA3 were traced in mapping software (Neurolucida Zeiss) using a 10x objective. The volume was determined by multiplying the surface area by the distance between sections (500 µm). Densities of BrdU+, EdU+, TRAP+, and c-Fos+ cells were estimated by multiplying the total number of labeled cells by 10. For immunohistochemical analysis, a subset of approximately 5 mice from the behavioral cohorts of ~15 mice were randomly selected. If the results exhibited high variability, one or two additional mice were included to improve data precision. Selection remained random throughout. This approach was chosen to balance the workload associated with immunohistochemistry while ensuring sufficient statistical power; given that behavioral experiments require larger sample sizes to achieve robust significance, only a subset could be included in this analysis.

## Distribution of TRAP+ and c-Fos+ cells in the DG

We followed methods presented in literature (*Perederiy et al., 2013*). Using stacks of images obtained from confocal microscopy, the localization of TRAP+ and c-Fos+ cells in the DG was assessed using Fiji software. *Distance from the subgranular zone (SGZ)*: The contours of each blade were manually outlined and the distance from the center of each cell body to the SGZ was manually measured. Cells located in the apex region were excluded from the counting.

*Distance from the apex*: Excluding the apex region, the size of each SB was measured as well as the distance of each cell from the apex. These distances were then expressed as a percentage of total blade length.

## Statistical analysis

Data are expressed as mean ± SEM. Statistical analyses were performed using Statview and Origin. Histochemical and other experimental data were assessed for normality using the Shapiro–Wilk test. For normally distributed datasets, comparisons between two groups were performed using unpaired *t*-tests, while one- or two-way ANOVAs were used for experiments with multiple factors, followed by appropriate post hoc tests (Tukey or Bonferroni) when significant. For datasets that deviated from normality, non-parametric tests (Mann–Whitney $U$ test or Kruskal–Wallis test) were applied, with Dunn's post hoc test and Bonferroni correction as appropriate. Differences were considered significant at $p \leq 0.05$.

## Acknowledgements

We thank members of the Contractor lab for helpful discussion and Ms. Astrid Castellanos for assistance during the performance of this study. This work was supported by NIH/NINDS 5R01NS115471 and NIH/NIMH R01MH130428 to AC.

## Additional information

### Funding

| Funder | Grant reference number | Author |
| --- | --- | --- |
| National Institutes of Health | R01NS115471 | Anis Contractor |
| National Institutes of Health | R01MH130428 | Anis Contractor |

The funders had no role in study design, data collection, and interpretation, or the decision to submit the work for publication.

## Author contributions

Charlotte CM Castillon, Conceptualization, Data curation, Formal analysis, Investigation, Methodology, Writing – original draft, Project administration; Shintaro Otsuka, Software, Investigation, Methodology, Writing - review and editing; John N Armstrong, Investigation, Methodology, Writing - review and editing; Anis Contractor, Conceptualization, Resources, Data curation, Formal analysis, Supervision, Funding acquisition, Visualization, Methodology, Writing – original draft, Project administration

## Author ORCIDs

Anis Contractor https://orcid.org/0000-0002-5131-2536

## Ethics

This study was performed in strict accordance with the recommendations in the Guide for the Care and Use of Laboratory Animals of the National Institutes of Health. All animal procedures were conducted in accordance with protocols approved by the Northwestern University Institutional Animal Care and Use Committee (IACUC) protocol #IS00015045.

Reviewer #2 (Public review): https://doi.org/10.7554/eLife.109611.4.sa1
Author response https://doi.org/10.7554/eLife.109611.4.sa2

---

# Additional files

## Supplementary files

MDAR checklist

## Data availability

All data needed to evaluate the conclusions in the paper are available in the manuscript and/or the Supplementary Materials. Source data for this study are openly available in Zenodo at https://doi.org/10.5281/zenodo.19209813.

The following dataset was generated:

| Author(s) | Year | Dataset title | Dataset URL | Database and Identifier |
|-----------|------|---------------|-------------|-------------------------|
| Castillon CCM, Otsuka S, Armstrong JN, Contractor A | 2026 | Subregional activity in the dentate gyrus is amplified during elevated cognitive demands | https://doi.org/10.5281/zenodo.19209813 | Zenodo, 10.5281/zenodo.19209813 |

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
