## [Editor Report · eLife Assessment]

This manuscript presents a **valuable** study of the activity and functional relevance of different circuits in the dentate gyrus of mice performing a pattern separation task. **Convincing** evidence is presented to support the paper's central conclusions. The study is likely to be of interest to those studying the subregional organization and cell type-specific functions of the dentate gyrus.

---

## [Referee Report · Reviewer #2 (Public review)]

In this study, the authors investigate how increasing cognitive demand shapes activity patterns in the dorsal dentate gyrus (DG). Using a touchscreen-based TUNL task combined with TRAP/c-Fos tagging, birth-dating of adult-born granule cells (abDGCs), and chemogenetic inhibition, they show that higher task demand increases mature granule cell (mGC) recruitment and enhances suprapyramidal (SB) versus infrapyramidal (IB) blade bias. Functionally, mGC inhibition reduces overall activity and impairs performance without disrupting blade bias, whereas inhibition of {less than or equal to}7-week-old abDGCs increases mGC activity, abolishes blade bias, and impairs discrimination under high-demand conditions. These findings suggest that effective pattern separation depends not only on overall DG activity levels but also on the spatial organization of recruited ensembles.

The integration of touchscreen TUNL with temporally controlled activity tagging and birth-dated cohorts is technically strong. Quantification of SB-IB bias and radial/apical distributions adds anatomical precision beyond bulk activity measures. The comparison between mGC and abDGC inhibition is conceptually compelling and supports dissociable functional roles. Overall, the data convincingly demonstrate that increasing cognitive demand amplifies blade-biased DG recruitment and that mGCs and abDGCs differentially contribute to both behavioral performance and network organization.

However, how abDGCs are integrated into the mGC network under high cognitive demand remains unresolved. Additional experiments are needed to clarify how abDGCs shape spatial recruitment patterns and whether they directly inhibit or indirectly regulate mGC activity to maintain high performance.

Furthermore, the authors frame "high cognitive demand" as a multidimensional construct encompassing broad behavioral challenge. It would strengthen the work to delineate how local abDGC-mGC circuit interactions regulate specific task components in real time. This will require higher temporal resolution approaches, as TRAP and c-Fos labeling integrate activity over prolonged windows and primarily reflect sustained engagement rather than moment-to-moment computations.

The central conclusion that dentate function depends on coordinated spatial recruitment rather than total activity magnitude is supported by the data, although mechanistic interpretations are tempered given methodological limitations.

Overall, this work advances models of adult neurogenesis by emphasizing a critical-period modulatory role of abDGCs in organizing DG network activity during high-demand discrimination. The combined behavioral and circuit-level framework is likely to be influential in the field.

Comments on revisions:

None remaining.

---

## [Author Response]

The following is the authors’ response to the previous reviews

**Public Reviews:**

**Reviewer #1 (Public review):**
This manuscript investigates how dentate gyrus (DG) granule cell subregions, specifically suprapyramidal (SB) and infrapyramidal (IB) blades, are differentially recruited during a high cognitive demand pattern separation task. The authors combine TRAP2 activity labeling, touchscreen-based TUNL behavior, and chemogenetic inhibition of adult-born dentate granule cells (abDGCs) or mature granule cells (mGCs) to dissect circuit contributions.This manuscript presents an interesting and well-designed investigation into DG activity patterns under varying cognitive demands and the role of abDGCs in shaping mGC activity. The integration of TRAP2-based activity labeling, chemogenetic manipulation, and behavioral assays provides valuable insight into DG subregional organization and functional recruitment. However, several methodological and quantitative issues limit the interpretability of the findings. Addressing the concerns below will greatly strengthen the rigor and clarity of the study.Major points:(1) Quantification methods for TRAP+ cells are not applied consistently across panels in Figure 1, making interpretation difficult. Specifically, Figure 1F reports TRAP+ mGCs as density, whereas Figure 1G reports TRAP+ abDGCs as a percentage, hindering direct comparison. Additionally, Figure 1H presents reactivation analysis only for mGCs; a parallel analysis for abDGCs is needed for comparison across cell types.

In Figure 1G and 1H we report TRAP+ abDGCs as a percentage rather than density because we are analyzing colocalization of the two markers, which are very sparse in this population. Given the very low number of double-labeled abDGCs, calculating density would not be practical. In the revised manuscript we have clarified the rationale for using these measures. As noted in the current text, we did not observe abDGCs co-expressing TRAP and c-Fos; we have made this point more explicit to guide interpretation of these data.

(2) The anatomical distribution of TRAP+ cells is different between low- and high-cognitive demand conditions (Figure 2). Are these sections from dorsal or ventral DG? Is this specific to dorsal DG, as itis preferentially involved in cognitive function? What happens in ventral DG?

The sections shown in Figure 2 were obtained from the dorsal dentate gyrus (see Methods, “Histology and imaging”: stereotaxic coordinates −1.20 to −2.30 mm relative to bregma, Paxinos atlas). From a feasibility standpoint, it is not possible to analyze the entire longitudinal extent of the hippocampus with these low-throughput histological approaches. We therefore focused on the dorsal DG, for which there is a strong functional rationale. A large body of work indicates that the dorsal hippocampus, and specifically the dorsal DG, is preferentially involved in spatial memory and in the fine contextual discrimination that underlies pattern separation. The dorsal hippocampus is critical for encoding and distinguishing similar spatial representations, a core component of the high-cognitive demand task used here. In contrast, the ventral DG is more strongly associated with emotional regulation and affective memory processing and is less implicated in high-resolution spatial encoding. For these reasons, the present study was designed to assess TRAP+ cell distributions specifically in the dorsal DG.

(3) The activity manipulation using chemogenetic inhibition of abDGCs in AsclCreER; hM4 mice was performed; however, because tamoxifen chow was administered for 4 or 7 weeks, the labeled abDGC population was not properly birth-dated. Instead, it consisted of a heterogeneous cohort of cells ranging from 0 to 5-7 weeks old. Thus, caution should be taken when interpreting these results, and the limitations of this approach should be acknowledged.

We agree that prolonged tamoxifen administration results in labeling a heterogeneous population of abDGCs spanning approximately 0 to 5–7 weeks of age, rather than a precisely birth-dated cohort. This is a limitation of this approach and we have included discussion of this in more detail in the revised manuscript.

(4) There is a major issue related to the quantification of the DREADD experiments in Figure 4, Figure 5, Figure 6, and Figure 7. The hM4 mouse line used in this study should be quantified using HA, rather than mCitrine, to reliably identify cells derived from the Ascl lineage. mCitrine expression in this mouse line is not specific to adult-born neurons (off-targets), and its expression does not accurately reflect hM4 expression.

We agree that mCitrine is not a marker that allows localization of hM4Di as it is well known that the mCitrine can be independently expressed in a Cre independent manner in this mouse. As suggested, we have removed the figure that showed the mCitrine and have performed immunohistochemical localization of the DREADD with an antibody against the HA tag. This is now shown in Figure 5.

(5) Key markers needed to assess the maturation state of abDGCs are missing from the quantification. Incorporating DCX and NeuN into the analysis would provide essential information about the developmental stage of these cells.

The goal of this study was to examine activity patterns of adult-born versus mature granule cells, rather than to assess maturation state. The adult-born neurons analyzed were 25–39 days old, an age at which point most cells have progressed beyond the DCX^+^ stage and are expected to express NeuN based on prior work. We therefore do not think that including DCX or NeuN quantification would provide additional information relevant to the aims or interpretation of this study.

Minor points:(1) The labeling (Distance from the hilus) in Figure 2B is misleading. Is that the same location as the subgranular zone (SGZ)? If so, it's better to use the term SGZ to avoid confusion.

We have updated Figure 2B, the Methods, and the main text to more explicitly localize this which it the boundary between the subgranular zone (SGZ) and the hilus.

(2) Cell number information is missing from Figures 2B and 2C; please include this data.

We have now added the cell number information to the figure legends. In Figures 2B and 2C, each point corresponds to a single cell, with an equal number of mice per group. The total number of TRAP^+^ cells per mouse is shown in Figure 1F, which reports TRAP^+^ cell densities by group.

(3) Sample DG images should clearly delineate the borders between the dentate gyrus and the hilus. In several images, this boundary is difficult to discern.

We made the DG-hilus boundaries clearer in the sample images to improve visualization and interpretation.

(4) In Figure 6, it is not clear how tamoxifen was administered to selectively inhibit the more mature 6-7-week-old abDGC population, nor how this paradigm differs from the chow-based approach. Please clarify the tamoxifen administration protocol and the rationale for its specificity.

We apologize for the confusion here. The protocol used in Figure 6 is the same tamoxifen chow–based approach as in Figure 5, differing only in the duration of tamoxifen exposure. Mice in Figure 5 received tamoxifen chow for 7 weeks, whereas mice in Figure 6 received it for 4 weeks, restricting labeling to a younger and narrower cohort of adult-born DGCs. Thus, the population targeted in Figure 6 is younger than that in Figure 5 and does not correspond to mature 6–7-week-old neurons. By contrast, the experiment in Figure 4 targets a more mature population, consisting predominantly of ~5-week-old adult-born neurons as well as mature granule cells, which are Dock10-positive and express Cre endogenously, allowing selective manipulation of this later-stage population.

We have corrected the paragraph accordingly and clarified the age range of the labeled populations in the revised manuscript.

Comments on revisions:I appreciate the authors' careful and thorough revisions. They have addressed all of my previous concerns satisfactorily, and the manuscript is now significantly strengthened. I have no further concerns.
**Reviewer #2 (Public review):**
In this study, the authors investigate how increasing cognitive demand shapes activity patterns in the dorsal dentate gyrus (DG). Using a touchscreen-based TUNL task combined with TRAP/c-Fos tagging, birth-dating of adult-born granule cells (abDGCs), and chemogenetic inhibition, they show that higher task demand increases mature granule cell (mGC) recruitment and enhances suprapyramidal (SB) versus infrapyramidal (IB) blade bias. Functionally, mGC inhibition reduces overall activity and impairs performance without disrupting blade bias, whereas inhibition of {less than or equal to}7-week-old abDGCs increases mGC activity, abolishes blade bias, and impairs discrimination under high-demand conditions. These findings suggest that effective pattern separation depends not only on overall DG activity levels but also on the spatial organization of recruited ensembles.The integration of touchscreen TUNL with temporally controlled activity tagging and birth-dated cohorts is technically strong. Quantification of SB-IB bias and radial/apical distributions adds anatomical precision beyond bulk activity measures. The comparison between mGC and abDGC inhibition is conceptually compelling and supports dissociable functional roles. Overall, the data convincingly demonstrate that increasing cognitive demand amplifies blade-biased DG recruitment and that mGCs and abDGCs differentially contribute to both behavioral performance and network organization.However, how abDGCs are integrated into the mGC network under high cognitive demand remains unresolved. Additional experiments are needed to clarify how abDGCs shape spatial recruitment patterns and whether they directly inhibit or indirectly regulate mGC activity to maintain high performance.Furthermore, the authors frame "high cognitive demand" as a multidimensional construct encompassing broad behavioral challenge. It would strengthen the work to delineate how local abDGC-mGC circuit interactions regulate specific task components in real time. This will require higher temporal resolution approaches, as TRAP and c-Fos labeling integrate activity over prolonged windows and primarily reflect sustained engagement rather than moment-to-moment computations.The central conclusion that dentate function depends on coordinated spatial recruitment rather than total activity magnitude is supported by the data, although mechanistic interpretations should be tempered given methodological limitations.Overall, this work advances models of adult neurogenesis by emphasizing a critical-period modulatory role of abDGCs in organizing DG network activity during high-demand discrimination. The combined behavioral and circuit-level framework is likely to be influential in the field.
**Reviewer #3 (Public review):**
This study examines the role of dentate gyrus neuronal populations, reflecting neurogenesis and anatomical location (suprapyramidal vs infrapyramidal blade), in a mnemonic discrimination task that taxes the pattern separation functions of the dentate. The authors measure dentate gyrus activity resulting from cognitive training and test whether adult neurogenesis is required for both the anatomical patterns of activity and performance in the cognitive task. The authors find that more cognitively challenging variants of the task evoked more dentate activity, but also distinct patterns of activity (more activity in the suprapyramidal blade, less in the infdrapyramidal blade). Using chemogenetic approaches they silence mature vs immature dentate gyrus neurons and find that only mature neurons (either the general population or specifically mature adult-born neurons), and not immature adult-born neurons, are required for the difficult version of the task. Inhibition of mature adult-born neurons furthermore increased overall activity in the dentate and reduced the biased pattern of activity across the blades, consistent with evidence that adult-born neurons broadly regulate dentate gyrus activity.Comments on revisions:I appreciate the efforts the authors have taken to revise this manuscript. I have only minor concerns with this revised version of the manuscript:Methods state that significance is defined as P<0.05 but some results are interpreted as significant when P=0.05. Either the alpha value needs to change or the interpretation needs to change.

We have corrected the statement in the Methods section to define statistical significance as *P ≤ 0.05*, which aligns with how significance was interpreted throughout the manuscript.

I believe the statistical results for group and blade effects for the ANOVAs, in Figs 2,3 & 4, appear to be switched (blade should be significant, not group).

We thank the reviewer for pointing out this mistake. We have corrected the reported statistical results for the group and blade effects in the manuscript accordingly.

I appreciate that sometimes there is not a perfect overlap between immunohistochemical signals, but I continue to believe that the spatially-non-overlapping TRAP and EDU signals in Fig 3 is caused by these 2 markers being in different cells. A Z-stack or orthogonal projection could verify/disprove this concern.

We agree that limited overlap in single optical sections can raise the possibility that TRAP and EdU signals originate from different cells. However, based on our imaging conditions and inspection across focal planes, the signals are consistent with being present within the same cells, with partial spatial separation likely reflecting subcellular localization and/or sectioning effects.